# Biological Functions of Hydrogen Sulfide in Plants

**DOI:** 10.3390/ijms232315107

**Published:** 2022-12-01

**Authors:** Zhifeng Yang, Xiaoyu Wang, Jianrong Feng, Shuhua Zhu

**Affiliations:** 1College of Chemistry and Material Science, Shandong Agricultural University, Tai’an 271018, China; 2Department of Horticulture, College of Agriculture, Shihezi University, Shihezi 832000, China

**Keywords:** hydrogen sulfide, resistance, cysteine residues, plant growth regulator, crosstalk, post-translational modification

## Abstract

Hydrogen sulfide (H_2_S), which is a gasotransmitter, can be biosynthesized and participates in various physiological and biochemical processes in plants. H_2_S also positively affects plants’ adaptation to abiotic stresses. Here, we summarize the specific ways in which H_2_S is endogenously synthesized and metabolized in plants, along with the agents and methods used for H_2_S research, and outline the progress of research on the regulation of H_2_S on plant metabolism and morphogenesis, abiotic stress tolerance, and the series of different post-translational modifications (PTMs) in which H_2_S is involved, to provide a reference for future research on the mechanism of H_2_S action.

## 1. Introduction

Hydrogen sulfide (H_2_S) is a colorless, flammable, toxic, and corrosive gas that gives rotten eggs their distinctive odor; it is involved in numerous physiological processes in the plant, such as seed germination [1], stomatal movement [2], root morphogenesis [3], photosynthesis [4], senescence and yield [5], and fruit ripening and quality [6]. One of the major targets of H_2_S signaling in eukaryotic cells is the mitochondria. In nature, sulfate (SO_4_^2−^)-reducing bacteria (SRB) use SO_4_^2−^ as a terminal electron acceptor, rather than oxygen (O_2_), to perform anaerobic respiration, resulting in the production of H_2_S as a major respiratory byproduct. They have a symbiotic relationship with aerobic sulfide-oxidizing bacteria, which can oxidize H_2_S back to SO_4_^2−^ and may, thus, be used continuously by SRB [7,8]. Initially, H_2_S was found to inhibit cytochrome c oxidase (COX) in mitochondria, and the research into H_2_S focused on its toxic effects [9]. Research into plant mitochondria has made some progress with the advent of deeper studies. Mitochondria are one of the compartments wherein H_2_S is produced [10]. At the subcellular level, H_2_S determines the ability of mitochondria to regulate energy production and prevent cellular senescence, thereby preventing leaf senescence in plants under drought-stress conditions [11]. H_2_S protects plants from salt stress by reducing hydrogen peroxide (H_2_O_2_) accumulation, regulating membrane stability and antioxidant systems in mitochondria [12]. These reports show that H_2_S has a positive effect on plant physiology.

On the other hand, H_2_S is involved in the defense mechanisms of plants against various abiotic stresses, including osmotic stress [13], drought stress [14], salt stress [15], extremely high- or low-temperature stress [16], and metalloids stress [17], by reducing reactive oxygen species (ROS) accumulation and actively mobilizing bioactive proteins, such as post-translational modification (PTM), as represented by *S*-sulfhydration [18,19]. H_2_S is capable of interacting with many plant hormones, such as abscisic acid (ABA) [20], auxin (AUX) [21], gibberellins (GAs) [22], ethylene (ETH) [6], salicylic acid (SA) [23], jasmonic acid (JA) [24], and brassinosteroids [25]. The H_2_S-mediated PTM targets are protein residues, particularly cysteine (Cys) residues. Structural changes in proteins can cause changes in protein activity. Subsequent changes in hormone signaling in which the protein is involved will then follow, for example, changes in respiratory burst oxidase homolog protein D (RBOHD) [26], Open Stomata 1 (OST1)/Sucrose nonferme-1(SNF1)-RELATED PROTEIN KINASE2.6 (SnRK2.6) [27], ABSCISIC ACID INSENSITIVE 4 (ABI4) [28], etc.

In this review, we describe H_2_S synthesis and the metabolic pathways, donors, and detection methods for H_2_S research, the specific functions of H_2_S in the plant life cycle, and regulation by H_2_S on tolerance against abiotic stresses. In addition, the progress of research on a range of different post-translational modifications/signaling systems regulated by H_2_S in plants is also summarized.

## 2. Biosynthesis of H_2_S in Organisms

### 2.1. Biosynthesis Pathway of H_2_S

The biosynthesis of H_2_S includes both non-enzymatic and enzymatic pathways.

#### 2.1.1. Non-Enzymatic Pathway

The non-enzymatic pathway of H_2_S biosynthesis occurs due to the reaction of thiols or thiol-containing compounds with other molecules [18,29,30,31]. Glutathione (GSH) reduces inorganic polysulfides or hydrolyzed inorganic sulfide salts, i.e., sodium sulfide (Na_2_S) or sodium hydrosulfide (NaHS) with water to produce H_2_S [29,30,31]. Cysteine is the preferred substrate for the non-enzymatic production of H_2_S, and the process is catalyzed by iron and vitamin B_6_ [32].

#### 2.1.2. Enzymatic Pathway

The trans-sulfuration pathway is the primary source of endogenous H_2_S and is the only way to produce endogenous cysteine, involving cellular sulfur metabolism and redox regulation [33] (Figure 1). Methionine, acting as a substrate, is catalyzed to produce cysteine eventually, with H_2_S as the byproduct. Cystathionine-β-synthase (CBS, EC 4.2.1.22), cystathionine-γ-lyase (CSE, EC 4.4.1.1), cysteine aminotransferase (CAT, EC 2.6.1.3), and 3-mercaptopyruvate sulfurtransferase (3-MST, EC 2.8.1.2) are all involved in this pathway. CBS and CSE are only located in the cytoplasm, while CAT and 3-MST can be present in the cytoplasm and mitochondria [34]. In the first irreversible step of converting methionine to cysteine, CBS catalyzes the condensation of homocysteine with serine (Ser) or cysteine to form cystathionine. The main role of CSE in the trans-sulfuration pathway is in the conversion of cystathionine to cysteine and α-ketobutyrate. CBS can produce H_2_S through β-substitution reactions [34,35,36]. Similarly, CSE can produce H_2_S via the β-elimination reaction with cysteine or the γ-replacement reaction between two homocysteine molecules. There is also a mechanism for H_2_S production, mediated by CAT and 3-MST in the transsulfuration pathway. CAT catalyzes l-cysteine and α-ketoglutarate to form 3-mercaptopyruvate (3-MP) and glutamate. The sulfur group of 3-MP is then transferred to 3-MST-accepting nucleophilic Cys247 in the presence of 3-MST, to produce 3-MST-bound persulfide and pyruvate. After this stage, the MST-persulfide reacts with thiols or is reduced by thioredoxin (Trx) to form H_2_S [37,38]. In addition, 3-MST transfers the sulfur group from 3-MP to cyanide, to form thiocyanate [39]. Similar to the CAT/3-MST pathway, there is also a d-amino acid oxidase (DAO, EC 1.4.3.3)/3-MST pathway to generate H_2_S (Figure 1). DAO metabolizes d-cysteine into 3-MP, which is metabolized into H_2_S by 3-MST [40].

In plants, the synthesis pathway of H_2_S can be divided into five types, according to the different substrates. These are cysteine degradation and sulfite reduction, cyanide detoxification, iron-sulfur cluster turnover, and carbonyl sulfide (COS) conversion [41] (Figure 1). Similar to animals, the metabolism of cysteine is the main source of endogenous H_2_S production in plants. The plants also have a characteristic sulfate reduction assimilation of H_2_S production. H_2_S synthesis occurs in chloroplasts, cytoplasm, and mitochondria [42]. First, H_2_S is mainly derived from cysteine degradation in the plant, catalyzed by different cysteine-degrading enzymes, including l-cysteine desulfhydrase (l-DES, EC 4.4.1.1), d-cysteine desulfhydrase (d-DES, EC 4.4.1.15), and l-3-cyanoalanine synthase (CAS, EC 4.4.1.9) [43]. Second, H_2_S is derived from the reductive assimilation of sulfite (SO_3_^2−^) in the plants (Figure 1). These two pathways of H_2_S synthesis are closely linked. Sulfate or atmospheric sulfur dioxide (SO_2_) is the source of SO_3_^2−^ production in the plant in the presence of adenosine 5′-phosphosulfate (APS) reductase (APSR, EC 1.8.99.2). Atmospheric SO_2_ can also produce SO_3_^2−^ spontaneously via non-enzymatic interaction with water. Sulfite reductase (SiR, EC 1.8.7.1) reduces SO_3_^2−^ to H_2_S in the presence of chloroplast enzymes and ferredoxin [44,45]. Under alkaline conditions in the chloroplast stroma, plants spontaneously transport HS^−^ (a dissociated form of H_2_S) into the cytoplasm (cytosol). With pyridoxal phosphate as a cofactor, l/d-cysteine is catalyzed by l/d-DES to produce pyruvate, NH_4_^+^, and H_2_S in the cytoplasm, chloroplasts, and mitochondria. Third, the nitrogenase Fe-S cluster-like (NFS/NifS-like) protein, which has similar activity to l-cysteine desulfurases, also catalyzes the conversion of cysteine to alanine and sulfur or sulfide using l-Cys as a substrate. It is also a possible source of H_2_S in plants [46,47]. The cyanide detoxification mechanism is also an important source of H_2_S in the plant (Figure 1). Ser and Acetyl-CoA are used to synthesize the intermediate reaction of *O*-acetyl-l-serine (OAS), catalyzed by Ser acetyltransferase (SAT, EC 2.2.1.30). *O*-acetyl-serine(thiol)lyase (OASTL, EC 2.5.1.47), also known as cysteine synthase, catalyzes the insertion of a particular sulfide (in this case, H_2_S) into the carbon skeleton via an elimination reaction, and produces cysteine [48]. CAS involves cyanide detoxification and regulates the production of H_2_S in mitochondria. H_2_S is both an intermediate reduction product of sulfate assimilation and a substrate for the synthesis of cysteine. The biosynthesis of cysteine in plastids implies a transition between a reduction in the assimilated sulfate-reducing pathway and actual metabolism [48]. The iron-sulfur cluster located in *Arabidopsis* mitochondria is capable of assembling the NIF system and presents cysteine desulfurase activity, which may also offer a potential source of H_2_S [49]. Besides the four H_2_S synthesis pathways described above, carbonic anhydrase (CA, EC 4.2.1.1) catalyzes the hydrolysis of COS to produce carbon dioxide (CO_2_) and H_2_S. The plants absorb COS from the air and achieve the efficient use of sulfur assimilation via CA [50,51], which may also be a source of endogenous H_2_S in the plant.

### 2.2. Metabolic Pathways of H_2_S

H_2_S exhibits diverse physiological and signaling roles, mainly in four distinct biochemical ways (Figure 2): (1) reacting with reactive molecule species, such as ROS, reactive nitrogen species (RNS), hypochlorite (HOCl), and reactive carbonyl species (RCS) [52,53,54]; (2) binding to the metal center of metalloproteins or the reduction of the hemoglobin center [55,56,57]; (3) post-translationally modifying proteins with specific structures (e.g., proteins containing cysteine residues (-SH)), which are mainly via *S*-sulfhydration [18,26,28,58]. The other PTMs are described in detail below [18]; (4) activities involving the oxidative and methylation pathways [33,59].

H_2_S reacts with the reactive molecule species, including ROS and RNS [18] (Figure 2). H_2_S can react with several biological oxidants, including superoxide radicals (O_2_^•−^), hydrogen peroxide (H_2_O_2_), hydroxyl radicals (HO·), nitrogen dioxide (NO_2_), peroxynitrite (ONOOH), and many others. H_2_S reacts readily with HOCl to form polysulfides (-S-S_n_-S-) [52]. Excessive ROS and RNS levels lead to oxidative stress when plants are exposed to adversity. In turn, H_2_S significantly impacts the products of the plant’s defensive system. Nitric oxide (NO) is an important signaling molecule. H_2_S can react with NO, leading to the formation of various nitrogen (nitrous oxide (N_2_O), nitroxyl (HNO), *S*-nitrosothiols (RS-NO/SNO), and sulfur derivatives (e.g., S^0^, S^−^), which are thus involved in physiological signaling. NO converts the adversity-induced O_2_^•−^ to the less toxic ONOO^−^. H_2_S further reacts with ONOO^−^ to form thionitrate (HSNO_2_) [54]. This property of H_2_S to actively scavenge ONOO^−^ provides strong support for the inference that it synergizes with NO to reduce ROS oxidative stress. In addition, H_2_S can react directly with NO to produce HNO and also react with RS-NO to form thionitrous acid (HSNO) [54]. HSNO can be metabolized to provide NO^+^, NO, and NO species, thus acting as a transportable NO reservoir in the organism that is involved in NO signaling.

H_2_S binds to the metal centers of metalloproteins or participates in electron transfer [18] (Figure 2). H_2_S involves the physiological regulation of the oxidative phosphorylation of the electron transfer chain (ETC) by means of binding to components of the ETC. It is mainly the direct binding of H_2_S to COX that affects ETC function, and the reduction of COX by H_2_S leads to the formation of HS^•^ or S^•−^ (which can interact with protein sulfhydryl groups (thiol)), affecting other components of the ETC. ETC complex IV, also known as COX, consists of two redox centers, Cyt a, Cu_A_, and Cyt a_3_, Cu_B_. H_2_S associates with the COX component, hematoxylin a_3_ (heme a_3_), and the Cu_B_ center, thus participating in the electron transfer of the ETC [55,56]. H_2_S favors the formation of a polar environment (tyrosine (Tyr) residues and Cu_B_ centers) around the heme a_3_ subunit, while H_2_S promotes heme a_3_ reduction to achieve an increase in COX enzyme activity at low concentrations. At high concentrations, H_2_S can bind directly to the component a_3_ and Cu_B_ centers of COX, resulting in the formation of the stable H_2_S-Cu_B_ and unstable hemoglobin H_2_S-Fe^2+^ inhibitory groups. In this case, the stability of the H_2_S-Fe^2+^ group is dependent on H_2_S concentration. However, the inhibition of COX by H_2_S can behave differently, depending on the concentration. Unlike medium concentrations, a high concentration of H_2_S is accompanied by the formation of stable hemoglobin a_3_ H_2_S-Fe^3+^ inhibitory groups, the inhibitory effect of which is irreversible [55,56,57]. In addition, the reduction of COX by H_2_S promotes increased ATP synthesis (which can bypass complex III to promote ETC activity) and the accumulation of reactive sulfur [33].

H_2_S can post-translationally modify proteins by converting the thiol group of cysteine residues to the persulfide group (-SSH) [18,26,28,58] (Figure 2). This modification is named *S*-sulfhydration [18]. The increased nucleophilicity of the converted persulfides, compared to the thiol group, highlights the highly reactive nature of *S*-sulfhydration. This also explains the potential for persulfides to act as mediators of sulfide signaling.

The metabolic pathways of H_2_S include oxidation and methylation [59] (Figure 2). The oxidation of H_2_S occurs in the mitochondria and involves several enzymes, such as sulfide quinone reductase (SQR) and the ethylmalonic encephalopathy 1 protein (ETHE1, also known as persulfide dioxygenase), thiosulfate sulfurtransferase (TST, also known as rhodanese), and mitochondrial sulfite oxidase. SQR oxidizes H_2_S in the inner mitochondrial membrane to produce persulfide species (e.g., glutathiol (GSSG)). At the same time, electrons released by SQR are captured by ubiquinone and transferred from H_2_S to coenzyme Q and to ETC at complex III. The persulfide is further oxidized by ETHE1 to produce SO_3_^2−^, which is further oxidized by sulfite oxidase to SO_4_^2−^, or by TST to S_2_O_3_^2−^ [59]. The metabolism of H_2_S by methylation occurs more as a complementary mechanism to oxidation and takes place in the cytoplasm. Then, thiol *S*-methyltransferase converts H_2_S to methanethiol (CH_4_S) and dimethyl sulfide (CH_3_)_2_S, which is further oxidized by rhodanese to produce thiocynate and SO_4_^2−^ [33].

## 3. H_2_S Donors, Inhibitors (Including Scavengers), and Detection Methods

### 3.1. Selection of Pharmacological Agents for H_2_S Pharmacology Experiments

At present, studies on H_2_S are based on the use of the donors and/or inhibitors (scavengers) of H_2_S. Typical H_2_S donors include Na_2_S, NaHS, morpholin-4-ium 4-methoxyphenyl (morpholino) phosphinodithioate (GYY4137), l-thiovaline (TV), and thioglycine (Figure 3) [14,60,61]. Notably, the specific targeting of mitochondrial H_2_S donors, including 10-oxo-10-(4-(3-thioxo-3H-1,2-dithiol-5yl) phenoxy) decyl) triphenylphosphonium bromide (AP39), 10-(4-carbamothioylphenoxy)-10-oxodecyl) triphenylphosphonium bromide (AP123), and RT01, have received much attention recently [62]. Unlike the acting mechanism of the traditional H_2_S donors, AP39 (a mitochondria-specific H_2_S donor) plays a direct role. The mitochondrial effects of Na_2_S are dependent on the NO/cyclic guanosine monophosphate (cGMP)/cGMP-dependent protein kinase (PKG) pathway. The inhibition of NO synthesis prevents the mitigating effects of Na_2_S and TV regarding cellular oxidative damage. This means that AP39 can be used for research even if upstream intracellular signaling (e.g., the NO/cGMP/PKG signaling pathway) is blocked due to environmental changes [60]. In terms of the mitigation of oxidative damage by H_2_S donors, their mitigation capacity can be listed in descending order of strength: AP39, TV, Na_2_S, and GYY4137 [60]. In addition, other characteristic H_2_S donors have been identified (Figure 3). Dialkyldithiophosphate is a new environmentally friendly H_2_S-releasing agent with 3 advantages. Firstly, it slowly degrades to release H_2_S. Secondly, the degradation products are natural, non-toxic, and can be used as fertilizer. Thirdly, only 0.28 mg of H_2_S is released when 1 mg of this substance is added to the soil near the seeds, avoiding growth inhibition at high concentrations [63]. Exogenous H_2_S inhibitors that act in the opposite direction include potassium pyruvate (PP), dl-propargylglycine (PAG), AOA, and hydroxylamine (HA) (Figure 3). Similar effects are reported with the use of exogenous H_2_S scavengers, such as hypotaurine (HT) (Figure 3) [64].

### 3.2. H_2_S Detection Methods

Depending on the purpose of the experiment, many methods measuring H_2_S have been developed successively, including photometric methods, gas chromatography, electrochemical methods, and biotransformation analysis [65,66]. The difference between these approaches is that proteomics studies need to consider the need to maintain protein function, while organelle studies focus more on in vivo-specific localization.

Fluorescent probes are also important in detecting H_2_S, including intracellular and in vitro detection. The former is mainly used for the in vivo imaging of H_2_S, while the latter is mostly used for biological sample detection. The selection of fluorescent probes should take into account their response rate and selection specificity. The ratiometric fluorescent probe eliminates most of the environmental factors’ interference by the ratio of fluorescence intensity at two wavelengths (i.e., self-correction), thus enabling the quantitative detection of the species under test when the probe concentration is unknown. Based on these advantages, the technique is considered to be an accurate method of measuring substances [67]. A large number of ratiometric fluorescent probes have been reported for the determination of H_2_S, e.g., the Cy-N_3_ H_2_S probe [68], the CouMC H_2_S probe [67], the coumarin-merocyanine dyad (CPC) H_2_S probe [69], the ratio-H_2_S 1/2 probe [70], and AcHS-2 ratiometric two-photon fluorescent probes [71]. In addition, the use of targeted mitochondrial H_2_S donors greatly simplifies studies at the subcellular level, such as those into AP39 [72], AP123 [73], and RT01 [62].

### 3.3. Detection of Protein S-Sulfhydration Modifications by H_2_S

Studies of H_2_S-mediated *S*-sulfhydration proteins are currently being carried out, following the biotin conversion approach as the basic process. The primary process has two steps involving the closure of the thiol-blocking reagent and the persulfide by an electrophile reagent and the release of the persulfide by a reducing agent. Specifically, there are four methods commonly used for *S*-sulfhydration testing. The first method is the classic electrophile *S*-methyl methanethiosulfonate (MMTS) method. MMTS is first used as a sulfhydryl blocker, followed by the labeling of persulfides with N-[6-(biotinamido) hexyl]-3′-(2′-pyridyldithio) propionamide (biotin-HPDP). This method includes *S*-nitrosation, in addition to *S*-sulfhydration labeling [74]. In the second method, iodoacetic acid (IAA) accurately detects the presence of sulfhydryl groups in the peroxidized proteins. The target groups (–SH and –SSH) in the protein are alkylated by IAA, thus achieving the blockage of the free thiol with protein persulfides. DTT specifically cleaves the -SSH group in the protein that is alkylated. Iodoacetamide-linked biotin (IAP) is labeled against the cleaved moiety by dithiothreitol (DTT). This method effectively eliminates the effect of intramolecular disulfides and intermolecular disulfides [75]. The third method, using N-ethyl maleimide, is the one that is most widely and commercially used. N-ethyl maleimide, which is linked to both the target groups (-SH and -SSH) in the protein, is used as a sulfhydryl group sealer. DTT specifically cleaves the target moiety (-SSH), which has been released by linking Cy5-conjugated maleimide to the site. The in-gel fluorescence signal is reduced in the samples containing persulfide [76]. The fourth method is the tag-switch test (tag-switch): a two-step reaction involving sulfhydryl blocking (SH-blocking (BR)) and tag-switching reagent substitution. The tag-switching reagent contains a reporting molecule (R) and a nucleophile (Nu). Nucleophiles differ in their reactivity toward persulfide adducts, due to their reactive properties. The selection of suitable nucleophilic reagents for this property allows for the specific detection of disulfide bonds in persulfides. For example, in the first step, methylsulfonyl benzothiazole (MSBT) is used as a sulfhydryl-blocking substance for SH-blocking (BR). Next, labeling is completed by binding to labeled tagged cyanoacetate derivatives, which effectively label the persulfides [77]. Recently, a new method for selective persulfide detection, named the dimedone-switch method, has been developed, which has the advantages of being specific, rapid, and stable [78]. The method is based on using dimedone probes for chemoselective persulfuration labeling. Since protein persulfides (PSSH) are very reactive, their reactivity is similar to that of cysteine residues. Therefore, it becomes difficult and important to design tools for selective labeling. The dimedone-switch method is roughly divided into two steps. In the first step, 4-chloro-7-nitrobenzofurazan (NBF-Cl)-treated material was used to block and detect thiols, amines, and sulfenic acids. Then, -S-S-signal detection is conducted, based on the dimedone/dimethyl ketone probe. NBF-Cl not only reacts with PSSH and thiols but also blocks sulfenic acids. NBF-Cl is key to the specificity of this method for the detection of persulfides. Conventional methods rely on the closure of sulfhydryl groups and persulfides with electrophilic reagents, followed by the release of the latter by means of reducing agents. These methods require a great deal of attention and are tedious and complex [79].

## 4. H_2_S Is Involved in Plant Growth and Development

### 4.1. H_2_S and Germination

Seed germination is crucial for the initiation of the plant’s life cycle (Figure 4). H_2_S delays seed germination in a dose-dependent manner [80]. This regulation does not act directly but probably occurs via a cascade involving other substances. Exogenous H_2_O_2_ can enhance the promotion of seed germination [81], and H_2_S acts upstream of H_2_O_2_ in seed germination [82]. The regulation of seed germination by H_2_S may be achieved by the persulfide modification of key proteins (ABI4) [1]. H_2_S is also able to indirectly regulate the transcription of AOX genes, thereby regulating seed germination [83].

H_2_S may promote germination by mitigating the adverse effects of multiple stresses on seeds. Exogenous H_2_S enhances endosperm amylase activity in germinating seeds, effectively reducing the accumulation of malondialdehyde (MDA) and H_2_O_2_ and promoting seed germination [84]. The application of H_2_S donors alleviates high-temperature-induced tissue viability, reduces MDA accumulation, and improves germination rates [85]. Exogenous H_2_S can promote the NO-induced salt resistance pathway and enhance salt tolerance in alfalfa seeds under salt stress [86]. In summary, H_2_S has a potential regulatory role in relation to seed germination. Low concentrations of H_2_S promote seed germination, while high concentrations of H_2_S inhibit seed germination [1,81,82]. This may be related to the properties of H_2_S itself. H_2_S inhibits COX in mitochondria [9]. From the perspective of energy metabolism, H_2_S is positive in terms of reducing energy consumption. H_2_S-mediated *S*-sulfhydration is one of the most important safeguarding mechanisms to protect proteins from peroxidation. From this point of view, it may be beneficial for seed germination. Toxicity studies have been a prominent feature of H_2_S since its discovery. This may be the reason for showing the inhibition of seed germination in some of the studies.

### 4.2. H_2_S and Growth Development

#### 4.2.1. H_2_S and Root Development

The root system is an essential source of water and minerals for plants (Figure 4) [87]. The factors affecting root development can be divided into intrinsic regulation and extrinsic inter-rooting influences (Table 1).

H_2_S involves the intrinsic regulation of root development through H_2_S-NO-carbon monoxide (CO) crosstalk [88]. Changes in root morphology and the regulation of root development are accomplished through various signaling networks. AUX is a vital growth hormone that regulates root cell replication, root elongation, and root morphological changes. PIN3- and PIN7-mediated changes in auxin allocation are responsible for the bending of roots toward the horizontal side of high AUX. AUX induces cell expansion through the synergistic activity of small auxin-up RNAs (SAUR), *Arabidopsis* H^+^-ATPase (AHA), and cell-wall-modifying proteins [89]. NO is actively involved in the regulation of the root system. Exogenous NO leads to adventitious root (AR) formation by mediating the growth hormone response in cucumber [90], is involved in the induction of root tip elongation in *Lupinus luteus* [91], regulates the formation of ARs [92] and lateral roots (LRs) [93], may also influence root lignification and growth hormone-dependent cell cycle gene regulation [94,95]. NO may be indirectly involved in root regulation as a component of the relevant signaling pathway. For example, NO is a downstream signaling molecule for the induction of LR by CH_4_ [96,97]. Similar to NO, exogenous CO treatment can also promote LR and AR formation. Exogenous CO induces LR and AR formation and the use of ZnPPIX (a CO scavenger) can reverse this effect. NO-CO crosstalk regulates root development. CO induces the formation of LRs that may be mediated by the NO/NOS pathway, and NO may act downstream of CO signaling [98]. The heme oxygenase isoenzyme-1 (HO-1) pathway is the primary source of endogenous CO in plants. This means that the oxidation of heme as a substrate to CO, free iron (Fe^2+^), and biliverdin (BV) is catalyzed by HOs [88]. NO modulates the activity of HOs and the transcriptional level of *HO-1* in soybean [99]. Under both normal and stress conditions, H_2_S has a precise regulatory effect on the primary, lateral, and adventitious roots of plants [88]. H_2_S causes the inhibition of filamentous actin (F-actin) bundles through the modification of F-actin by *S*-sulfhydration, which leads to the inhibition of root hair growth [88]. There are other mechanisms by which H_2_S regulates root polarity transport in plants through its effect on F-actin. Actin is essential for the transport and distribution of PIN proteins. Actin-binding proteins (ABPs) act as the downstream effectors of H_2_S signaling, thereby regulating the assembly and depolymerization of F-actin in root cells. H_2_S inhibits growth hormone transport by altering the polar subcellular distribution of PIN proteins, thereby regulating root development [100]. H_2_S-NO crosstalk regulates root growth and development. Numerous biological compounds are produced by H_2_S (nucleophilic) and NO (electrophilic), via oxidation or nitrosation reactions, and these substances are also actively involved in physiological regulation. H_2_S can be involved in physiological regulation through a cascade of signals, in common with NO [101]. Exogenous NaHS alleviates the adverse effects of salt-stress-induced oxidative damage in *Medicago sativa* roots, while the application of c-PTIO (a NO scavenger) reverses these effects [86].

H_2_S regulates root development by actively participating in signaling events to plant–bacteria symbiosis [102,103,104,105]. Aqueous solutions of H_2_S with a weakly acidic pH may affect the activity of inter-rooted microorganisms. Recent studies have also found that exogenous H_2_S can modify the expression of target genes and achieve an improved root structure by mediating the microRNA (miRNA) [15].

In addition, root growth is promoted by H_2_S at low concentrations and inhibited by H_2_S at high concentrations, which is typical of the effect of H_2_S on root development [106,107]. High concentrations of H_2_S regulate the root system architecture (RSA) by affecting the transport of growth hormones [106]. High concentrations of H_2_S activated the ROS-(MITOGEN-ACTIVATED PROTEIN KINASE 6) MPK6-NO signaling pathway and inhibited primary root (PR) growth. During this process, ROS production, activated by exogenous H_2_S, is required for NO generation, and MPK6 mediates H_2_S-induced NO production [107].

#### 4.2.2. H_2_S and Photosynthesis Photomorphogenesis

Photosynthesis is the process by which plants convert light energy into chemical energy. H_2_S involves plant photosynthesis by regulating the redox state, the non-stomatal factors, and stomatal movement [27,108,109,110].

Both the photoreaction centers and photosystem II (PSII) reaction centers are largely affected by redox states [111]. Exogenous H_2_S enhances the scavenging of ROS, regulating the activities of antioxidant enzymes (superoxide dismutase (SOD), ascorbate peroxidase (APX), and peroxidase (POD), catalase, inhibiting the accumulation of H_2_O_2_ and MDA, and improving the stress tolerance of the plant [108,109,112]. H_2_S maintains high guaiacol peroxidase (GPX), POD, APX, catalase, and GSH reductase (GR) activities and down-regulates the chlorophyll degradation-related genes, *BoSGR*, *BoCLH2*, *BoPaO*, and *BoRCCR* in broccoli [113].

The amount and state of the relevant components of photosynthesis (non-stomatal factors) are also limiting factors for photosynthesis. Exogenous H_2_S alleviates the net photosynthetic rate (*P*_n_), stomatal conductance (*G*_s_), intercellular CO_2_ concentration (*C*_i_), transpiration rate (*E*), the maximal quantum yield of PSII photochemistry (*F*_v_/*F*_m_), the effective quantum yield of PSII photochemistry (*Φ*_PSII_), the photochemical burst of ribulose-1,5-bisphosphate ribulose carboxylase, and the decrease in electron transport rate (*ETR*) in low light-stress (LL)-induced tall fescue seedlings [110]. Further analysis reveals that *P*_n_ follows an opposite trend to *C*_i_, implying that the high level of photochemical efficiency maintained by H_2_S may be dominated by non-stomatal factors [110]. NaHS effectively increases chlorophyll content and ribulose1,5-bisphosphate carboxylase (RuBISCO) activity [114].

H_2_S regulates photosynthesis by participating in stomatal movement. In *Arabidopsis*, H_2_S is involved in stomatal closure through induction of the NO-mediated 8-nitro-cGMP/8-mercapto-cGMP synthesis pathway (one of the classical mechanisms of NO-induced stomatal closure). Conversely, the application of NO scavengers and NO-producing enzyme inhibitors can inhibit H_2_S-induced stomatal closure. H_2_S may be upstream of the NO-induced stomatal closure pathway [115]. ROS changes are early marker events for the stomatal closure response. ROS are important signaling molecules involved in stomatal movement [116]. H_2_S triggers the production of ROS via NADPH oxidase (NADPHox) and phospholipase D (PLD) [117]. Crosstalk between H_2_S and other signaling molecules (ABA, NO, ROS) is the leading method of stomatal movement. H_2_S-mediated post-translational modification, SnRK2.6, by *S*-sulfhydration is a novel regulatory mechanism for ABA signaling and ABA-induced stomatal closure [27]. In addition, the H_2_S-mediated modification of 1-aminocyclopropane-1-carboxylic acid (ACC) oxidase (ACO) by *S*-sulfhydration becomes a way of regulating ABA stomatal regulation [118]. Modifying target proteins by H_2_S *S*-sulfhydration regulates stomatal movement and is described in Section 5.3.

Stomatal movement is also closely related to the redox state of photosynthetic components. The redox state of the PSII receptor, lateral plastid quinone (PQ), may be transmitted to the stomatal guard cells in leaves through signal transduction, thereby regulating the stomatal opening and closing and affecting the stomatal conductance [119]. The above findings suggest that H_2_S can promote the onset of photosynthesis and the expression of photosynthetic enzymes in plants. The reason for this is presumed to be that H_2_S induces the expression of genes involved in photosynthetic electron transfer via thiol redox modification [114].

### 4.3. H_2_S and Maturity

#### 4.3.1. H_2_S and Flowering

Flowering is a sign that the plant has changed from nutritional to reproductive growth (Figure 4). Suppressor of Overexpression of Constant 1 (*SOC 1*), Flowering Locus T (*FT*), and Flowering Locus C (*FLC*) are essential genes for the regulation of flowering. FLC is a MADS-box transcription factor. FLC achieves the regulation of the vernalization pathway by controlling *SOC 1* and strongly represses the expression of genes that promote plant flowering. Exogenous 100 μmol L^−1^ H_2_S promotes flowering in heading Chinese cabbage via *S*-sulfhydration, by weakening or eliminating the binding capacity of BraFLCs to the downstream promoters [120]. It suggests that H_2_S is actively involved in the flowering process of plants.

#### 4.3.2. H_2_S and Fruit Ripening

Ripening and completion are two critical stages in the fruit life cycle (Figure 4). From ripening to completion, a series of changes occur within the fruit: starch is converted into sugar, organic acids are broken down, the acidity decreases, the pectinase activity increases, causing pectin differentiation, and the pulp becomes soft. ETH is an essential factor in the regulation of fruit-ripening initiation [121,122].

H_2_S maintains fruit quality and delays fruit senescence and decay by antagonizing the effects of ETH [6]. Exogenous H_2_S maintains high levels of nutrient-related metabolites (e.g., anthocyanins, starch, soluble proteins, ascorbic acid, flavonoids, total phenols, etc.) in fruit, enabling the maintenance of fruit quality during storage [6,123,124]. Inhibiting ETH synthesis and signal transduction are crucial in how H_2_S is involved in the fruit-ripening process. H_2_S inhibits the expression of ETH biosynthesis (*MdACS1*, *MdACS3*, *MdACO1*, and *MdACO2*) and signal transduction (*MdETR1*, *MdERS1*, *MdERS2*, *MdERF3*, *MdERF4*, and *MdERF5*)-related genes in apples [124]. In addition, ETH induces the production of H_2_S in guard cells during osmotic stress; ETH-induced H_2_S negatively regulates ETH biosynthesis through the *S*-sulfhydration of LeACO [118]. These findings suggest that there is a crosstalk between H_2_S and ETH.

The H_2_S-induced attenuation of respiration is a critical way of delaying fruit senescence. Mitochondria are the place where respiration occurs and the leading site of ROS production [125]. The lipoxygenase (LOX, EC l.13.11.12) catalyzes and mediates the oxylipin synthesis pathway, also known as the fatty acid pathway, the primary method by which fruit aroma substances are formed. The enzymes downstream of LOX, hydroperoxide lyase (HPL), allene oxide synthase, alcohol dehydrogenase (ADH), and alcohol acyltransferases (AAT), are critical in the synthesis of alcohols, aldehydes, esters, and other derivatives in plants [126]. The excessive accumulation of ROS induces autophagy and programmed cell death (PCD) in plant cells (this is not described in detail here, but as described in Section 5.3). Decreased LOX activity and reduced ROS production are specific manifestations of reduced respiration. Improving the ability to scavenge ROS is essential for plants, offering a way to slow down senescence [127]. It is also a strategy for H_2_S to slow down ripening by maintaining high levels of antioxidant enzymes, such as APX, catalase, GPX, POD, and GR, and by reducing the activities of LOX, polyphenol oxidase (PPO), phenylalanine ammonia-lyase (PAL), and protease in a dose-dependent manner, inhibiting the post-harvest senescence and decay of the fruit [113,124,128].

H_2_S can also regulate post-harvest ripening and decay by directly affecting the degradation of the cell wall [129,130]. The cell wall is an essential structure for maintaining plant morphology and defense against fungi. Pectin methylesterase (PME, EC 3.1.1.11), polygalacturonase (PG, EC 3.2.1.15), and endo-β-1,4-glucanase (EGase, EC 3.2.1.4) are crucial factors influencing the process of post-harvest cell wall softening [121]. Chitinase (CHI, EC 3.2.1.14) and beta-1,3-glucanase (GNS, EC 3.2.1.6) are critical factors in fruit decay [129]. In strawberry plants, exogenous H_2_S significantly inhibited fruit softening by decreasing the activities of PME, PG, and EGase [129]. Similarly, in the Chilean strawberry, 0.2 mmol L^−1^ NaHS could reduce the expression of pectin degradation-related genes (*FcPG1*, *FcPL1*, *FcEXP2*, *FcXTH1*, *FcEG1*) and alleviate pectin degradation, thereby prolonging fruit shelf-life [130].

### 4.4. H_2_S and Senescence

Plant senescence is the process of the decline in life activity that leads to natural plant death. In this process, the plant transports stored material or even disintegrated protoplasm to new organs or other organs, facilitating the survival of the individual plant or the continuation of the species [131]. From this point of view, senescence is not a purely passive and reactive process; instead, there is precise regulation via antagonistic regulatory substances in the plant [132]. The following section on plant senescence refers specifically to leaf senescence.

The antagonistic regulation of PCD by H_2_S and GAs provides a precise regulatory mechanism for seed germination. The PCD of the pasteurized layer cells contributes to the maintenance of the stability of the internal environment [133]. During mid-seed development, the embryo-surrounding region (ESR) is degraded to provide space for the growing embryo [134]. GAs positively regulate PCD in the cells of the pasteurized layer. It has been reported that GAs induce ROS production via NADPHox in barley and that ROS promote the expression of *GAMYB* in dextrin cells, which, in turn, induces the expression of amylase [135,136]. Nucleases and proteases that are found downstream of ROS are associated with dextrin vesiculation and cell death [137]. ROS may not directly control the PCD of the paste powder layer, but instead, feed back the GAs signal by disrupting the SLN1. GAs affects the breakdown of starchy endosperm (SE) by targeting the Della protein, SLENDER1 (SLN1), for degradation and by inducing the expression of secreted amylase [136]. Unlike the action of GAs, exogenous H_2_S delays PCD and enhances the secretion of alpha-amylase [138]. In addition, GAs inhibits l-DES activity in dextrin cells, thereby reducing the synthesis of endogenous H_2_S [139].

H_2_S regulates plant senescence through cascade crosstalk with other plant hormones and signaling molecules [5,140]. Exogenous H_2_S inhibits the yellowing process by down-regulating the transcription of chlorophyll-degrading genes (*BoSGR*, *BoCLH2*, *BoPaO*, and *BoRCCR*) and maintaining high levels of chlorophyll, carotenoids, anthocyanins, and ascorbic acid in leaves [113]. It is well known that ETH is involved in the regulation of the senescence process in plants and that it mainly plays a role in accelerating maturation and senescence [141]. Leaf abscission is an important physiological phenomenon of senescence. H_2_S plays an antagonistic role in ETH-promoted petiole abscission [142]. NO also inhibits ETH synthesis in a dose-dependent manner [143]. The ETH-NO-H_2_S signaling pathway becomes an important pathway for regulating plant maturation.

Furthermore, the nuclear factor-E2-related factor (Nrf2) is the primary regulator of the antioxidant response in cells, whereas the Kelch-like ECH-associated protein 1 (Keap1) is a negative regulator of Nrf2. H_2_S is post-translationally modification of (Keap1), induces Keap1 to segregate Nrf2, enhances Nrf2 nuclear translocation, and stimulates Nrf2-targeted downstream genes to prevent senescence [144]. H_2_S also prevents senescence by regulating the signaling of the oxidative stress pathways.

**Table 1 ijms-23-15107-t001:** H_2_S is involved in plant growth and development.

Development	Plant Species	H_2_S Doses	Effects	References
Germination	*Arabidopsis thaliana*	12 μmol L^−1^ NaHS	Activated AOX Mediated cyanide-resistant respiration pathway	[83]
	*Arabidopsis thaliana*	0.1 mmol L^−1^ NaHS	Maintained the protein stability of ABSCISIC ACID-INSENSITIVE 4	[1]
Root development	Peach (*Prunus persica* (L.) Batsch)	0.2 mmol L^−1^ NaHS	Increased the concentration of endogenous auxin Upregulated the expression of *LATERAL ORGAN BOUNDARIES DOMAIN 16*	[145]
	Cucumber (*Cucumis sativus* L.)	100 μmol L^−1^ NaHS	Promoted the occurrence of adventitious roots Regulated osmotic substance content (proline) and enhanced antioxidant ability	[106]
Photosynthesis	Wheat (*Triticum aestivum* L.)	200 µmol L^−1^ NaHS	Increased photosynthesis and carbohydrate metabolism	[109]
	Garlic (*Allium sativum*)	200 µmol L^−1^ NaHS	Increased the values of net photosynthetic rate, transpiration rate and stomatal conductance	[108]
Flowering	Heading Chinese cabbage (*B. rapa* L. syn. *B. campestris* L. ssp. *chinensis* Makino var. *pekinensis* (Rupr.) J. Cao et Sh. Cao)	100 µmol L^−1^ NaHS	Promoted plant flowering by weakening or eliminating the binding abilities of BraFLCs to downstream promoters via *S*-sulfhydration	[120]
Maturity and senescence	Strawberry (*Fragaria chiloensis* (L.) Mill.)	0.2 mmol L^−1^ NaHS	Maintained fruit firmness Delayed pectin degradation Downregulated the expression of polygalacturonase, pectate lyase, and expansin	[130]
	Goji berry (*Lycium barbarum* L.)	1.4 mmol L^−1^ NaHS	Improved postharvest quality Increased bioactive compounds accumulation Boosted antioxidant capacity	[128]

NaHS: sodium hydrosulfide; AOX: alternative oxidase.

## 5. H_2_S Improves Plant Stress Tolerance

Unlike animals, the ability to tolerate the environment is crucial for plants. Even if the factors of the environment are different, the same adverse effects can be induced. Abiotic stresses, such as salinity, low temperature, and metalloids induce the onset of oxidative and osmotic stress, causing the accumulation of ROS in plants, the onset of membrane lipid peroxidation, impaired cell membrane stability, and impaired root and leaf function, leading to the onset of cellular water loss and photo-inhibition in plants. H_2_S can counteract the adverse effects of abiotic stress by regulating several critical physiological and biochemical processes (Figure 5). Among these, the main functions induced by H_2_S are typically to up-regulate antioxidant metabolic activity, accumulate osmotic regulation protective substances, and enhance stomatal movement/non-stomatal photosynthesis [14,146] (Table 2).

### 5.1. H_2_S and Oxidative Stress

Exposure of plants to various abiotic stresses, such as drought, salinity, temperature extremes, toxic metals, and hypoxia, causes an imbalance in endogenous redox homeostasis or oxidative stress. The excessive accumulation of ROS, such as O_2_^•−^ and H_2_O_2_, is the first step in the onset of oxidative stress, triggering autophagy and PCD [134]; it also causes protein oxidation, membrane lipid peroxidation, and the subsequent blockage of membrane-related physiological processes [147].

Changes in ROS levels in cells can alter the structure and function of a variety of proteins, thereby affecting many different signal transduction pathways. Under stress, interactions between ROS production and scavenging in the different compartments of plants can generate specificity [148]. Among them, peroxisomes are subcellular compartments that are important for ROS metabolism in plants. In plant peroxisomes, xanthine oxidoreductase (XOR) activity produces uric acid, which is accompanied by the production of O_2_^•−^. Nitric oxide synthase (NOS)-like enzyme generates NO by using l-arginine and Ca^2+^ in substrate activity. NO reacts with O_2_^•−^, thereby converting it to the less toxic ONOO^−^ [149]. H_2_O_2_-producing enzymes (Catalase, GOX) are regulated by NO-mediated *S*-nitrosation and tyrosine nitration [150]. O_2_^•−^ can also be converted to H_2_O_2_ by CuZnSOD. In addition, glycolate oxidase (GOX), acyl-CoA oxidase (AOX), urate oxidase (UO), polyamine oxidase, copperamine oxidase (CuAO), sulfite oxidase (SO), and sarcosine oxidase (SOX) are important sources of the peroxisomal H_2_O_2_ pool. In the peroxisome, catalase (located in the substrate) works with APX (located in the membrane) to break down H_2_O_2_ [149]. In plants, both catalase [151] and GOX [152] activities can be inhibited by H_2_S. Studies have found that H_2_S is present in plant peroxisomes [151]. Evidently, H_2_S and NO play an important role in the metabolism of ROS in the peroxisome. Antioxidant enzymes also play an important role in ROS scavenging. Proteomics studies have shown that antioxidant enzymes are important targets of H_2_S-mediated *S*-sulfhydration, including matrix sAPX and cystoid thylakoid tAPX, catalase 1/2/3, DHAR1/2/3, GR1/2, etc. [153]. Chen et al. speculated that H_2_S may interact with ROS homeostasis by acting against oxidases and/or other targets in the signaling pathway [154]. From the above examples, it is clear that H_2_S is involved in all ROS metabolism processes.

The interaction between H_2_S and other signaling molecules jointly regulates ROS. Both NO and H_2_S enhance plant antioxidant capacity by reducing the excess production of ROS, decreasing lipid peroxidation, and increasing the activity of major antioxidant enzymes [155]. The pharmacological experiments showed that H_2_S donors increased NO content, and vice versa, accompanied by the upregulation of several antioxidant enzyme activities (e.g., SOD, catalase, and APX) [156]. In maize, 0.2 mmol L^−1^ SNP (a NO donor) promotes the activity of endogenous H_2_S synthases (l-DES, CAS, and CS) and H_2_S accumulation [157]. Similarly, in the tomato, NO was observed to stimulate H_2_S accumulation by upregulating the transcript levels of H_2_S biosynthetic enzymes and acting downstream of NO to reduce oxidative stress [158]. Exogenous H_2_S reduced NO levels and controlled salinity-induced oxidative and nitrosative cell damage [159]. Carbon monoxide (CO) has been shown to protect plants from the harmful effects of excess ROS [160]. Heme oxygenase-1 (HO-1) is an important source of endogenous CO in plants [161]. In pepper, NaHS induces the *CsHO-1* gene and CsHO-1 protein expression in a time-dependent manner [162]. Hemoglobin (CO scavenger) treatment enhanced H_2_S biosynthesis (l-cysteine desulfhydrase, l-DES), which induced the accumulation of H_2_S in tobacco cells [163].

### 5.2. H_2_S and Osmotic Stress

Plants grow in a sedentary manner and therefore need to respond and adapt to harsh environmental conditions to survive. Similar to oxidative stress, osmotic stress can be induced by numerous abiotic stresses. The immediate consequence of osmotic stress in plants is water deficit. Osmoregulation is an essential response made by plants undergoing osmotic stress, which is manifested by the massive accumulation of osmoprotective substances, such as proline (Pro), soluble sugars (glucose, sucrose, and alginate), glycine betaine (GB), glycine (Gly), polyamines (PAs), and sugar alcohols. The accumulation of these osmoprotective substances contributes to the plant’s ability to resist stress. Among them, Pro is the typical osmoprotective substance, the accumulation of which mitigates the adverse effects of abiotic stresses. The physiological functions of this substance are shown in four aspects: relieving physiological dehydration and maintaining plasma membrane stability; forming hydrocolloids to protect protein molecules; effectively removing ROS; offering an effective means of energy storage under abiotic stress [164,165].

H_2_S acts as a downstream signaling molecule that regulates stomatal closure, actively participating in stomatal closure and reducing water dissipation, thereby improving tolerance to osmotic stress. PLD is a significant lipid hydrolase that hydrolyses the membrane phospholipids to produce phosphatidic acid (PA) [166]. PLD-δ is required for ABA-mediated stomatal closure and is essential for stabilizing the membrane system and balancing osmotic pressure when plants experience hyperosmotic stress [167]. H_2_S is located downstream of PLD-δ and promotes stomatal closure in response to osmotic stress [168], alleviating osmotic stress by increasing PLD and inhibiting the ROS [169]. In addition, the H_2_-H_2_S pathway has been found to regulate the stomata. In *Arabidopsis*, H_2_S is involved in osmotic stress tolerance as a downstream molecule of H_2_-regulated stomatal closure [170]. In cereals, osmotic stress induces the production of endogenous H_2_S, the accumulation of which enhances osmotic stress tolerance by mediating DNA methylation [13]. ACO is a key enzyme in ETH biosynthesis in plants. When tomato plants suffer osmotic stress, ACO is activated, and ETH is produced in large quantities. In one study, the application of 200 μmol L^−1^ of NaHS and ETH both induced stomatal closure and water retention. An HT supply counteracted the effects of ETH or osmotic stress on stomatal closure. On the other hand, ETH-induced H_2_S negatively regulates ETH biosynthesis via the *S*-sulfhydration of LeACO1/2. H_2_S also indirectly inhibited the transcription of *LeACO1* and *LeACO2*. This showed that endogenous H_2_S was downstream of osmotic stress signaling. H_2_S can be involved in osmotic stress signaling through either direct or indirect means [118].

### 5.3. H_2_S and Drought Stress

Drought stress is followed by a range of adverse plant effects, including water deficit, osmotic stress, oxidative stress, impaired cell integrity, and damage to PS II [14].

H_2_S promotes the accumulation of osmoprotective substances and alleviates the adverse effects of osmotic stress (induced by drought stress) [171,172] (Figure 5). Water deficit is the result of a combination of root water uptake rate and leaf transpiration rate, which is essentially caused by osmotic stress. The typical response to water deficit in plants is osmoregulation, mainly in the form of osmoprotective substances such as Pro, soluble sugars (glucose, sucrose, and alginate), GB, Gly, PAs, sugar alcohols, etc. The accumulation of these osmoprotective substances contributes to improving plant stress tolerance. During drought stress, exogenous H_2_S promotes the accumulation of Pro, GB, and trehalose in wheat [173], increases the levels of soluble sugars, PAs, Pro, and betaine in spinach seedlings, and up-regulates several genes related to PAs and soluble sugar biosynthesis, enhancing drought tolerance [174].

H_2_S mitigates damage from drought stress by regulating stomatal movement [175] (Figure 5). The ABA response is the classic response mechanism for stomatal movement [176,177]. Under drought stress, l-DES catalyzes the degradation of cysteine as an essential pathway for H_2_S production. After NaHS treatment, ABA-induced stomatal closure is attenuated in *lcd* mutants, and the transcript expression of ABA receptor candidates is up-regulated; in *aba3* and *abi1* mutants, stomatal density is reduced, and l-DES expression and H_2_S synthesis are decreased [178]. Drought stress-induced *DES1* expression is abolished in the *aba3* mutant. The addition of ABA or the expression of *ABA3* or *DES1* to the guard cells of *aba3*/*des1* double-mutants cannot alter the wilting phenotype of these mutants. However, with the application of the NaHS treatment, the wild-type phenotype is completely restored [179]. These findings suggest that H_2_S is involved in ABA-mediated stomatal movement via crosstalk with ABA. A growing number of reports have revealed its mechanisms. In guard cells, ABA regulates the activities of several ion channels in promoting stomatal closure and inhibiting stomatal opening, leading to changes in guard cell expansion pressure and stomatal closure, which is the central modulation of stomatal movement [180]. H_2_S may be an essential link in the regulation of stomata by ABA through ion channels, while influencing the expression of ABA receptor candidates and thereby regulating stomatal closure in guard cells [178]. H_2_S can also be involved in ABA-mediated stomatal movement, via the regulation of MPK4, to alleviate the water deficit caused by drought stress. In the guard cells, ROS acts as a second messenger for ABA-regulated stomatal closure signals [181]. In *Arabidopsis thaliana*, ROS production by the NADPHox pathway (e.g., *Arabidopsis thaliana* respiratory burst oxidase homolog protein D and F (*AtROHD* and *AtROHF*)) is the rate-limiting link in ABA signal transduction [182]. H_2_S increases the endogenous H_2_O_2_ in guard cells by affecting NADPHox activity and enhancing its ability to produce ROS [117]. This effect of H_2_S may be related to the transcriptional activation of NADPHox respiratory burst oxidase homolog (*RBOH*) and the modification of RBOH by *S*-sulfhydration [26]. Sucrose nonferme-1(SNF1)-RELATED PROTEIN KINASE2 (*SnRK2*), a significant switch for downstream ABA signaling in guard cells, can be activated by ABA [183]. ABA-induced endogenous H_2_S is involved in stomatal closure via peroxide-modified SnRK2 activity and improves drought tolerance [27]. In *Arabidopsis thaliana*, endogenous H_2_S induces the opening of the K^+^ channel (the outward K^+^-channel), which acts as the main osmoregulatory channel in response to drought stress, causing K^+^ efflux and Ca^2+^ and Cl^−^ efflux, leading to stomatal closure [180]. With the activation of the S-type anion channels (SLAC1), anions efflux out, playing a vital role in stomatal closure [184]. Exogenous H_2_S activates SLAC1, which occurs due to dependence on SnRK2.6, and elevates the intracellular free Ca^2+^ levels in *Arabidopsis* [185]. In addition, H_2_S enhances drought resistance by means of ABA non-dependent stomatal closure. In tobacco plants, H_2_S-induced stomatal closure in the guard cells is caused by the specific inactivation of inward-rectifying K^+^ channels (I_KIN_) [186]. The above finding suggests that a complex signaling network of ABA, H_2_O_2,_ and H_2_S cascades provides the regulatory mechanism for stomatal closure under drought stress, and that stomatal movement is accompanied by changes in the flux of several ions across the plasma membrane, carried by ion channels. In addition, in *Arabidopsis*, ABI4 acts downstream of l-cysteine desulfurase1 (DES1) to control the ABA response. The *S*-sulfhydration of ABI4 occurs at Cys250, triggered by ABA in a time-dependent manner, while the loss of function of DES1 weakens this process [28].

H_2_S alleviates drought stress-induced photosynthetic decline. Drought stress directly inhibits chlorophyll synthesis, reduces PSII-related light-trapping pigments, and decreases ribulose-1,5-bisphosphate carboxylase oxygenase (Rubisco) activity [187]. Exogenous H_2_S promotes the rapid conversion of the D1 protein and thereby restores PSII activity [188]. Aquaporins (AQP) are channels in the biological membranes that facilitate the movement of water across membranes, and their status and activities are key factors affecting the ability of plants to cope with drought stress. In addition, stomatal movement under drought stress provides a guarantee for the smooth progress of photosynthesis. Drought stress closes stomata and reduces *C*_i_, leading to an inhibition of the photosynthetic carbon reduction cycle, presenting a hindrance to photosynthesis [189]. This is potentially also the reason for the high photosynthetic capacity maintained by exogenous H_2_S under drought stress.

Exogenous H_2_S increases water channel protein expression (*SoPIP1;2*) in spinach seedlings [174], thereby improving drought resistance. The mechanism of the AQP channel opening and closing is regulated by PTMs, such as phosphorylation, methylation, acetylation, deamidation, and glycosylation. H_2_S is a potential source of these PTMs, but the exact mechanism of the regulation of AQP remains to be further elucidated [190].

### 5.4. H_2_S and Saline Stress

H_2_S maintains intracellular homeostasis when plants suffer salt stress (Figure 5). Salt stress mainly disrupts the thermodynamic balance of water and ions in the plant cells, leading to osmotic stress, ionic imbalance, and ionic toxicity [191,192]. The maintenance of intracellular homeostasis (ionic and osmotic balance) in plants is one of the most important strategies to counteract the adverse effects of salt stress. In barley, H_2_S is able to maintain low Na^+^ levels in the cytoplasm by enhancing the transcript levels of PM H^+^-ATPase (*HvHA1*) and the Na^+^/H^+^ reverse transporter protein (*HvSOS1*) to achieve Na^+^/H^+^ homeostasis, which is mediated by NO signaling [193]. Studies in *Cyclocarya paliurus* also found that H_2_S alleviated the adverse effects induced by salt stress in close correlation with NO levels [194]. The stomatal limitation is caused by salt stress reducing the water content in the leaves, triggering the closure of stomata, which leads to a decrease in *C*_i_ and a reduction in the plant’s photosynthetic rate. Exogenous H_2_S upregulates the expression of *PM H^+^-ATPase*, *SOS1*, and *SKOR* so as to maintain Na^+^ and K^+^ homeostasis and alleviate salt-stress-induced stomatal limitation-induced reductions in the photosynthetic properties, chlorophyll fluorescence, and stomatal parameters in cucumber plants [195]. In mutants lacking H_2_S and JA, stomatal density and stomatal index values are increased, while exogenous NaHS reverses this phenomenon [196]. H_2_S inhibits ETH synthesis by inhibiting the activity of ACO via *S*-sulfhydration [118]. It has also been found that the MT tolerance of iron deficiency (ID) and salt stress may be related to downstream signaling crosstalk between NO and H_2_S [197].

### 5.5. H_2_S and Extreme Temperature Stress

Extreme temperature is an important factor limiting plant growth and productivity. Unlike animals, plants are more likely to only tolerate temperature extremes passively. Improving their tolerance to temperature extremes is, therefore, even more critical. Depending on the temperature range in which the stress occurs, extreme temperature stress can be classified into freezing stress, chilling stress, and heat stress [198,199]. The effects of extreme temperatures on plant physiology have commonalities, including altered membrane fluidity, cellular water loss, protein denaturation, disruption of the RNA secondary structure, and, in particular, photo-inhibition and impaired metabolic homeostasis [200]. It is important to note that the protein denaturation caused by high temperatures is often irreversible. Freezing stress is accompanied by mechanical damage from ice crystals [198,201].

H_2_S improves plant resistance to low-temperature stress (Figure 5). Chilling stress promotes the activity of l-DES and d-DES in grapes, resulting in a significant increase in H_2_S content, suggesting that this may be a protective response by the plant to chilling stress [202]. The transcriptional response to cold stress can be divided into two phases: early and late. The early genes that respond mainly encode the transcription factors, while the late genes that respond belong to a core group of cold-inducible genes. The early encoded transcription factors, such as the C-repeat binding factor (*CBF*) or dehydration response element-binding factor 1 (*DREB1s*), are essential. Late cold-induced gene products that are directly involved in stress protection and the maintenance of cellular homeostasis include cold-regulated (*COR)*, cold-induced (*KIN)*, low-temperature-induced (*LTI)*, responsive to dehydration (*RD)*, and late embryo genesis-abundant (*LEA)* products [203]. These molecules and proteins, including ions, lipids, protein kinases, and transcription factors, are interconnected and form the basis of the plant’s response to cold stress. H_2_S upregulates cold stress-related mitogen-activated protein kinase (MAPK), especially MPK4, in *Arabidopsis* in response to cold stress by mediating the regulation of *ICE1*, *CBF3*, *COR15A*, and *COR15B* by MPK4, while inhibiting stomatal opening [204]. H_2_S can also directly influence *S*-sulfhydration MPK4 and increase MPK4 kinase activity in response to cold stress [205]. The effect of H_2_S on low-temperature stress is multifaceted. The exposure of cucumber root systems to chilling stress results in H_2_S promoting the expression of plasma membrane proton pump isoforms (*CsHA2*, *CsH4*, *CsH8*, *CsH9*, and *CsHA10*). The modulation of ATPase activity by H_2_S is more pronounced than that by NO and H_2_O_2_ [206]. The effects of exogenous NO, NO scavengers, and NO synthesis inhibitors on H_2_S metabolism differ during the low-temperature storage of peaches, implying that the regulation of endogenous H_2_S metabolism by exogenous NO is not a simple linear regulation [207]. NaHS alleviates plant photo-inhibition and membrane damage induced by exposure to chilling stress [208]. H_2_S may cause this effect by triggering downstream signals (e.g., GSH) regarding its chilling tolerance [209]. H_2_S also upregulated the expression of antioxidant enzyme genes (*CaSOD*, *CaPOD*, *CaCAT*, *CaAPX*, *CaGR*, *CaDHAR*, and *CaMDHAR*) to achieve improved cold tolerance [210].

In terms of heat stress, H_2_S crosstalks with signaling molecules (GTs) [211] and plant hormones [212,213]. In tobacco plants, CO improves cellular heat resistance, while NaHS enhances CO-induced heat resistance but can be attenuated by PAG, a specific inhibitor of H_2_S biosynthesis, or its scavenger, HT [163]. The application of NaHS and GYY4137 enhances SNP-induced heat tolerance; H_2_S may be a downstream signaling molecule for NO-induced heat tolerance in maize seedlings [214]. Heat stress activates H_2_S biosynthesis in plants and induces H_2_S accumulation. *S*-nitrosoglutathione reductase (GSNOR) is a key enzyme associated with the NO cycle in plants. H_2_S locates upstream of GSNOR and effectively eliminates RNS and ROS by upregulating GSNOR activity [215]. GSNOR may be a vital component linking NO to H_2_S. High temperature-induced photo-inhibition can be mitigated well by NO with H_2_S [211]. SA is a hormone that is involved in plant growth and development and that actively participates in plant heat tolerance. In maize (*Zea mays* L.), PAG and HT attenuate SA-induced heat tolerance. SA increases l-DES activity, which induces the accumulation of endogenous H_2_S, implying a positive role for SA and H_2_S crosstalk in plant heat tolerance [212]. Cellular water deficit, caused by heat stress, is a significant cause of injury; ABA-regulated stomatal closure is vital for managing water loss in leaves. In tobacco plants, ABA upregulates l-DES activity and induces the accumulation of endogenous H_2_S. ABA-induced heat resistance is enhanced by NaHS and is conversely weakened by PAG and HT [213]. In addition, exogenous H_2_S induces heat-shock proteins (HSP70, HSP80, and HSP90) and water channel proteins (PIP), while activating a coordinated network, associated with heat shock defense at the transcriptional level, that favor the formation of protective molecules [216].

### 5.6. H_2_S and Metalloids Stress

Metalloids, e.g., nickel (Ni), molybdenum (Mo), zinc (Zn), copper (Cu), chromium (Cr), lead (Pb), and cadmium (Cd) in soil are not entirely harmful; plant micronutrients (e.g., Ni, Zn, Mo, Cu, and Fe) are not toxic at low concentrations [217]. At specific concentrations, metalloids, represented by Ni, can significantly prevent the plants’ uptake and use of crucial mineral elements [218]. Conversely, some of the metallic elements, such as Cr, lead (Pb), and Cd, and mercury (Hg), are toxic whenever they are present [217]. The exposure of plants to metalloids stress induces the endogenous production of H_2_S [219]. Once generated, H_2_S can move freely across the plant membrane as a form of resistance to metalloids stress [220].

The mechanism by which H_2_S affects plant tolerance to metalloids is similar to oxidative stress and osmotic stress. H_2_S enhances the antioxidant system (via the upregulation of POD, SOD, catalase, and APX) to counteract oxidative stress (via the downregulation of MDA, H_2_O_2_, and EL) [17,221] and improves physiological processes, such as photosynthesis (Figure 5) [222]. The difference is that cell-wall metabolism is an integral part of H_2_S resistance to metalloids stress. The plant cell wall is an important barrier to the movement of metalloids across the membrane [223]; metal-tolerant proteins, represented by metallothionein and phytochelatin, are vital in maintaining cation homeostasis in the plant cell wall [224,225]. Recently, it has been found that exogenous H_2_S stimulates endogenous metal-binding protein activity, which, in turn, alters the accumulation of Cd in the cell wall, thereby reducing the mobility of intracellular metal ions [225].

In addition, the crosstalks between H_2_S and other signaling molecules (e.g., GTs), plant hormones, and plant growth regulators (PGRs) are involved in the related stress responses. H_2_S crosstalks with other GTs (NO, CO, CH_4_) to mitigate the damage to HMs [222]. In particular, H_2_S and NO both have similar defense responses: increased antioxidant metabolism, inhibition of the accumulation of HMs, and the triggering of Ca^2+^ [222]. Exogenous NO induces endogenous H_2_S, enhances reactive oxygen metabolism and photosynthesis, and eliminates the adverse effects of Cr (VI) on tomato seedlings [226]. Ca^2+^ is the second messenger of cellular resistance signaling. The calcium-dependent protein kinases (CDPKs) are important components of Ca^2+^ signaling, and these CDPKs are Ca^2+^-dependent. CDPKs are upstream of H_2_S and can be modulated to improve Cd stress tolerance in plants [227]. SA is a phenolic compound that has a positive effect on plant stress tolerance. The exogenous administration of SA with H_2_S triggers NO signaling in vivo while increasing these osmoregulatory substances alleviates the toxic effects of lead stress. H_2_S-SA crosstalk favors the expression of ASA-GSH metabolic activity [228]. Methyl jasmonate (MeJA) alleviates Cd stress-induced damage and enhances the expression of homeostasis-related genes (*MTP1*, *MTP12*, *CAX2*, and *ZIP4*), which process is mediated by H_2_S [24]. Thiamine (vitamin B1) is an important coenzyme that is involved in many metabolic pathways, especially in those processes related to energy metabolism (carbon assimilation and respiration) and is considered a plant growth regulator that is involved in plant responses to adversity [229]. H_2_S and NO are involved in THI-induced tolerance to Cd toxicity [230].

**Table 2 ijms-23-15107-t002:** H_2_S improves plant stress tolerance.

Plant Species	Stressors	H_2_S Doses	Protective Effects	References
*Arabidopsis thaliana*	*Oxidative*	0.5 mmol L^−1^ NaHS	Repressed glycolate oxidase activities	[152]
*Arabidopsis thaliana*	Osmotic	150 mmol L^−1^ NaHS	Involved in osmotic stress-triggered stomatal closure	[168]
Safflower	Drought: 70 and 50% field capacity	0.5 and 1.0 mmol L^−1^ NaHS	Increased the accumulation of secondary metabolites Strengthened the antioxidant capacity Regulated elemental uptake	[171]
Wheat (*Triticum aestivum* L.)	Drought: 30% field capacity	10 mg m^−3^ SO_2_	Triggered proline accumulation Activated antioxidant enzymes Changed expression level of transcription factors Increased H_2_S content	[172]
*Cyclocarya paliurus*	Salinity: 100 mmol L^−1^ NaCl	0.5 mmol L^−1^ NaHS	Maintained chlorophyll fluorescence Regulating nitric oxide level Improved antioxidant capacity	[194]
Wheat (*Triticum aestivum* L.)	Heat: 40 °C	200 µmol L^−1^ NaHS	Reduced glucose sensitivity Increased the activities of SOD, catalase, and the AsA-GSH cycle	[211]
Pepper (*Capsicum annuum* L.)	Chilling: 10 °C/5 °C day/night	1 mmol L^−1^ NaHS	Enhanced the antioxidant capacity Increased the enzyme transcription levels Reduced the contents of O_2_^•−^, H_2_O_2_, and MDA	[210]
Wheat (*Triticum aestivum* L.) Rice (*Oryza sativa* L. var.)	Metalloids: 20 μmol L^−1^ Cr(VI)	15 μmol L^−1^ NaHS	Maintained fruit firmness Delayed pectin degradation Downregulated the expression of polygalacturonase, pectate lyase, and expansin	[221]

SOD: Superoxide dismutase; AsA-GSH: ascorbate-glutathione cycle; MDA: malondialdehyde; O_2_^•−^: superoxide radical; H_2_O_2_: hydrogen peroxide.

## 6. H_2_S Regulates a Range of Different PTMs/Signaling Systems

The PTM of proteins offers an important pathway by which to exert biological effects related to life activities, environmental responses, and epigenetics. Emerging evidence suggests that H_2_S exerts its function by regulating various PTMs. Here, some of these recent developments are highlighted.

### 6.1. H_2_S Regulates the Post-Translational Modification of Protein Cysteine Residues (R-SH)

The common oxidative post-translation modifications (oxPTMs) of protein cysteine sulfhydryl groups include *S*-sulfhydration, *S*-sulfenylation, and *S*-nitrosylation. H_2_S is directly or indirectly involved in these oxPTMs [231,232] (Figure 6). The exposure of plants to stress causes the accumulation of ROS and RNS, which ultimately leads to oxidative stress. Originally, cysteine thiols may undergo different oxPTMs, for example, *S*-nitrosylation and *S*-sulfenylation. The products of the above two modifications are mostly oxidized compounds that can be reduced in cells via reducing agents (GSH, thioredoxin, and glutaredoxin) [233]. *S*-sulfhydration has a protective effect on peroxidation. Upon a further increase in stress, the protein will react with ROS/RNS and form an adduct (RSSO_3_H) that can be restored to free thiols by thioredoxin [231]. In *Arabidopsis* and pea, the Cys32 residue, as exemplified by peroxidase, can be targeted by two PTMs (*S*-nitrosylation and *S*-sulfhydration) [234,235]. Therefore, the regulatory effect of H_2_S on oxPTMs is complex and nonlinear. Chemically speaking, *S*-sulfhydration usually increases the reactivity of the target protein, while *S*-nitrosylation usually decreases the protein activity [236].

#### 6.1.1. H_2_S and *S*-Sulfhydration

*S*-sulfhydration (persulfidation) involves the modification of protein cysteine residues to form persulfides (cysteine thiols (RSH) are mercapturised to mercaptothiols (RSSH)), which results in the production of persulfide groups (-SSH) [237]. The synthesis of persulfides results from both the enzymatic and direct non-enzymatic production of persulfides from H_2_S. The primary enzymatic sources of persulfides are: (1) formations catalyzed by mercaptopyruvate sulfurtransferase (MST); (2) the sulfur oxidation pathway, in which the persulfide product is produced in a two-step reaction. Sulfide-quinone oxidoreductases (SQRs) catalyze the oxidation of H_2_S to sulfane sulfur, which remains covalently attached to the enzyme. This sulfane sulfur can then be transferred to the sulfite to form thiosulfate [238]. (3) Formed in the enzymatic cleavage of cystine by CBS or CSE, H_2_S is directly and non-enzymatically generated as a peroxisulfide, a precursor to highly reactive sulfur-containing compounds, including sulfenic acids, *S*-nitrosated cysteines, pre-existing inter- or other inter- or intramolecular disulfides, or via the facilitation of the formation of persulfides from H_2_S and protein sulfhydryl groups in the presence of metal ions. In addition, persulfides can be used as carriers of sulfane sulfur and participate in the “trans-*S*-sulfhydration” reaction.

*S*-sulfhydration is one crucial way in which H_2_S is involved in life activities [139] (Figure 6). In eukaryotes, cellular autophagy is a highly conserved mechanism of material degradation (the ability to spontaneously eliminate cellular components) [239]. This mechanism is of great significance for differentiation, development, and cell survival. Autophagy-related proteins (ATG) are a group of critical proteins that are involved in the autophagic process. The H_2_S target, ATG, offers the most robust evidence for its direct involvement in cellular autophagy. In animals, H_2_S modifies the Cys150 residue of GAPDH by *S*-sulfhydration, to achieve the regulation of cellular autophagy. Similarly, in plants, H_2_S can modify the residues of Cys170 of ATG4 and Cys103 of ATG18 via *S*-sulfhydration to achieve the inhibition of cellular autophagy [239]. This mechanism of inhibition of autophagy is independent of redox conditions [240]. The energy sensor, Snf1-related protein kinase 1 (SnRK1), the kinase target of rapamycin (TOR), ATG1 kinase complex, and the endoplasmic reticulum stress sensor inositol-requiring enzyme-1 (IRE1) are autophagy regulators that have been identified in plants [241]. The *S*-sulfhydration-based modification of these target proteins by H_2_S is another important way in which they are involved in cellular autophagy [239]. ABA signaling is the classical mechanism that regulates stomatal movement [242]. Similarly, H_2_S modifies specific cysteine residues through *S*-sulfhydration, thereby affecting stomatal closure [26,28,175], for example, ABI4 [28] and RBOHD [26]. Notably, the complex signaling network of the H_2_S, ABA, and H_2_O_2_ cascade crosstalk is an important way to activate the stomatal closure regulatory mechanisms [27,117,181]. In addition, potential target proteins for H_2_S occur in redox homeostasis, energy status, and cytokinesis-related enzymes, including APX [235], catalase [151], RBOHD [26], glyceraldehyde 3-phosphate dehydrogenase (GAPDH) [235], NADP-isocitrate dehydrogenase (NADP-ICDH) [243], NADP-malic enzyme (NADP-ME) [244], actin [245], etc. This implies that H_2_S is involved in the above metabolic activities via *S*-sulfhydration-based modifications.

H_2_S cannot react directly with thiols but can react with oxidized cysteine residues [246]. *S*-sulfhydration requires H_2_O_2_ signaling via sulfenylation. H_2_S can react with oxidized cysteine residues to obtain sulfenic acid (R-SOH) or with protein nitrosothiols (R-SNO) to obtain protein persulfides, but the latter process is thermodynamically unfavorable [54]. It is evident that H_2_S-mediated *S*-sulfhydration does not exist completely independently. The study of specific protein targets for *S*-sulfhydration remains the subject of intense debate.

#### 6.1.2. H_2_S and *S*-Sulfenylation

The occurrence of H_2_S-mediated *S*-sulfhydration may be predicated on *S*-sulfenylation [19,236]. H_2_S finds it difficult to directly reduce the disulfide bonds inside the protein-forming S-sulfide, from the thermodynamic and kinetic points of view [247]. Thiols are easily oxidized, and the presence of intracellular ROS can react with free thiols to undergo *S*-*s*ulfenylation. *S*-*s*ulfenylation will then produce oxidized cysteine. The presence of this class of residues is a direct target to enable the action of H_2_S to occur [236]. H_2_S participates in the endogenous cycle of mercaptan (RSH), *S*-sulfenylation, and *S*-sulfhydration (Figure 6). When a protein containing a thiol group [242] is exposed to ROS, the thiol group is reversibly converted to sulfenic acid (RSOH), at which point the protein is considered to undergo *S*-sulfenylation. RSOH is highly reactive, scavenging ROS and converting them to disulfides. Firstly, the ROS are scavenged, and these ROS can further irreversibly oxidize RSOH to form sulfinic acid (RSO_2_H) or sulfonic acids (RSO_3_H). Secondly, RSOH can react with H_2_S to produce water and RSSH. Similarly, RSSH can scavenge ROS in two steps to form RSH (the initial product is an adduct (RSSO_3_H) and RSSO_3_H is dependent on Trx activity for the further reaction to form RSH), or it is dependent on Trx activity to achieve direct conversion back to RSH [231]. The propensity of cysteine residues to undergo oxidation is mainly influenced by three aspects: (1) thiol nucleophilicity; (2) the surrounding protein micro-environment; (3) the proximity of the target thiol to the ROS source [248]. Cysteine residues with low pKa content exist as thioanions under normal conditions; they are more susceptible to attack by various oxidants and are also susceptible to *S*-sulfhydrate [18].

#### 6.1.3. H_2_S and *S*-Nitrosylation

H_2_S is actively involved in the modulation of *S*-nitrosylation products. NO-mediated PTMs represent the primary mode of NO bioactivity, including tyrosine nitration [249], *S*-nitrosylation, and metal-nitrosylation [250], of which *S*-nitrosylation is the central part. *S*-nitrosylation is the specific linkage of NO to the protein thiol group [242] of a cysteine residue, to form SNO [251]. *S*-nitrosylation is not only closely related to the NO signaling molecules but is also actively involved in gene expression. Both NO and H_2_S lipophilic molecules have some important properties in common: they can diffuse to the cell membrane, react with thiol groups, and mediate two important PTMs, namely, *S*-nitrosation and *S*-sulfhydration. The thiol group of the cysteine residue associates H_2_S with NO. H_2_S influences NO production and its metabolites by affecting NO synthase, and NO alters the bioavailability of H_2_S by acting on H_2_S-producing enzymes [252,253,254]. *S*-nitrosylation is naturally affected as NO mediates the typical PTMs. H_2_S can react with the *S*-nitrosylated product SNO to form nitrosopersulfide (SSNO^−^), which is regarded as an intermediate in the crosstalk between NO and sulfide [255]. That is because SSNO^−^, spontaneously and multi-step reversibly, forms a variety of inorganic polysulphides, including HSNO (HNO and HSSNO, etc., are involved in the modification of cysteine residues). H_2_S reacts with SNO to form HSNO (the smallest of the RSNO), which acts as a NO carrier to diffuse freely between cells, entering the cell and promoting the nitrosylation of proteins [54]. H_2_S is reversible with HSNO to form HNO, and HNO mediates the synergistic effect of sulfide and NO on TRPA1 channel activation [256]. The pKa of HSSNO is approximately 5, implying the insensitivity of the molecule to a sulfhydryl-mediated reduction in cells (at a physiological pH) [257], which avoids reduction by effective enzyme systems in the organism (the NADPH-driven bio-reduction mechanisms of Trx and GR systems). SSNO-derived products can be decomposed to sulfane sulfur, so that SSNO^−^ becomes a stable carrier for sulfane sulfur delivery [255].

#### 6.1.4. H_2_S and *S*-Glutathionylation

Similar to the above modifications (*S*-sulfhydration, *S*-sulfenylation, and *S*-nitrosylation), *S*-glutathionylation is also a PTMs for the reaction of a cysteine residue (R-SH) with a specific substrate; the difference is that the substrate is GSH [232].

The deglutathionylation of proteins can be catalyzed by glutaredoxin (Grx) and Trx. *S*-glutathionylation is a defense mechanism to avoid the excessive oxidation of cysteine residues and is mainly dependent on the status of GSH/GSSG. *S*-glutathionylation is, thus, a regulator of the redox state of cells, particularly in mitochondria [232]. Under oxidative stress, glutathionylation protects Trx from ROS-induced irreversible oxidation [258]. H_2_S can also convert R-SSG to R-SSH [232]. When CBS (the key enzyme for endogenous H_2_S synthesis) is modified by *S*-glutathionylation, it enhances the synthesis of H_2_S [259]. H_2_S is important for maintaining cellular GSH and glucose homeostasis in C_2_C_12_ myotubes and enhances total protein *S*-glutathionylation [260]. The above study demonstrated that H_2_S interacts with *S*-glutathionylation. In addition, *S*-glutathionylation appears to mediate the inhibition induced by *S*-nitrosylation in the presence of GSNO in vitro, suggesting that *S*-sulfenylation is most likely part of the signaling involved in *S*-nitrosylation, but the exact manner of involvement is yet to be investigated at this time [261].

### 6.2. H_2_S and Phosphorylation

H_2_S-mediated protein phosphorylation regulates plant physiological activity. Protein phosphorylation is the most widespread PTM in living organisms, mainly through the action of protein kinases and phosphatases on specific amino acid residues (typically, the hydroxyl groups of Ser, Thr, and Tyr), adding or removing one or more phosphate groups, thereby effectively altering the structure and activity of the substrate protein [262]. The phosphorylation modifications of proteins are closely associated with a variety of biological processes. In *Arabidopsis thaliana*, PSM and hinge multisite auto-phosphorylation are key to regulating protein degradation and Pr-to-Pfr transformation [263]. Phytochromes (PHY) and cryptochromes (CRY) are important photoreceptors for plant photomorphogenesis. PHY regulatory effects are strongly influenced by R/FR light-induced phosphorylation. The CRY regulatory effect is strongly influenced by blue light-induced phosphorylation. Red light (RL) exposure showed the transient activation of cysteine desulfurase activity, the key enzyme for endogenous H_2_S, whereas white light (WL) or blue light (BL) exposure inhibited the activity of this enzyme. l-DES is phosphorylated after exposure to RL or BL. Therefore, light can modulate the production of H_2_S. This process relies on the phosphorylation modification of l-DES [264]. In the root system, the H^+^ gradient generated by PM H^+^-ATPase is an important driver for maintaining ion homeostasis by the plasma membrane Na^+^/H^+^ anti-porter (SOS1). Under salt stress, H_2_S regulates *PM H^+^-ATPase* gene expression and phosphorylation status [265]. CDPKs regulate cellular recognition and signal transduction through reversible protein phosphorylation and increase Cd tolerance in *Arabidopsis thaliana* by enhancing H_2_S signaling [227]. The rapid recovery of the D1 protein facilitates the alleviation of photo-inhibition. Under drought stress, H_2_S alleviates photo-inhibition by mediating the phosphorylation of the D1 protein and accelerating the D1 protein turnover [188]. *SnRK2.6* is predominantly expressed in the guard cells. SnRK2.6-mediated ABA-induced stomatal closure is an important stress response of plants in response to drought. Crosstalk exists between phosphorylation and *S*-sulfhydration. The *S*-sulfhydration of SnRK2.6 increased the phosphorylation level of S175 residues. The phosphorylation status of S267 affects the peroxisulfation of SnRK2.6 in vivo and in vitro. Phosphorylation promotes *S*-sulfhydration, which alters the structure and increases the activity of SnRK2.6, ultimately protecting the guard cells [266].

### 6.3. H_2_S and Ubiquitination (Ub)

Ub is one of the mechanisms of PTMs that is prevalent in living organisms; it adds ubiquitin to the substrate protein and thereby labels the target protein. Its primary biochemical function is to provide a marker for the subsequent selective degradation of the protein. The ubiquitin-proteasome system (UPS), of which Ub is the core component, is the main pathway for intracellular protein degradation and consists of Ub, ubiquitin-activating enzyme (E1), ubiquitin-conjugating enzyme (E2), ubiquitin-protein ligase (E3), the proteasome and its substrate (protein) [267].

H_2_S can be involved in regulating protein degradation in cells by modifying ubiquitinated and deubiquitinated components [40,268] (Figure 6). E1 uses the energy generated by ATP hydrolysis to create a thioester bond between the sulfhydryl group of its own catalytic site, cysteine, and the carboxyl group of ubiquitin. The activated ubiquitin is then transferred to the sulfhydryl group of E2. Finally, the transfer of ubiquitin to a specific substrate is carried out by E3. In the UPS, E3 confers specificity to protein degradation reactions. E3 ubiquitin ligases are divided into the homologous type and the E6-AP carboxyl terminus (HECT)-type, really interesting new gene (RING)-type, and RING-between-RING (RBR)-type [269]. In humans, exogenous H_2_S modifies ubiquitin-specific peptidase 8 (USP8) via *S*-sulfhydration to promote the USP8-mediated deubiquitination of parkin (an RBR-type E3 ubiquitin ligase) for the effective elimination of dysfunctional mitochondria [268]. Similarly, H_2_S regulates parkin activity and enhances its neuroprotective activity by incorporating bound sulfane sulfur into the cysteine residues [40].

### 6.4. H_2_S and Histone Acetylation

The covalent modification of acetyl groups (acetyl groups) from acetyl CoA to the ε-amino group of the N-terminal lysine residue of the histone molecule is called histone acetylation [270].

The mutually antagonistic activity of histone acetyltransferases (HATs) and histone deacetylases (HDACs) is an important way of regulating histone acetylation in plants. HATs catalyze the acetyl transfer of acetyl coenzyme A on histone lysine residues, to achieve acetylation [271]. Plant HATs are classified into four categories, including p300/CREB, TATA-binding protein-associated factors, the MYST family of proteins (MOZ, Ybf2/Sas3, Sas2, and Tip60), and the general control non-repressible 5 (Gcn5)-related N-acetyltransferases (GNATs) [272]. In contrast, HDACs remove the acetyl group from the histone tails and achieve deacetylation, leading to chromatin condensation and reduced gene expression activity [273]. Plant HDACs are divided into three categories, including reduced potassium dependency 3/histone deacetylase 1 (RDP3/HDA1), the silent information regulator 2 (SIR2), and histone deacetylase 2 (HD2) [270].

H_2_S can regulate cellular function by affecting the level of histone acetylation [274,275,276] (Figure 6). In animals, H_2_S can regulate cellular function by affecting the acetylation and deacetylation of histones. For example, H_2_S upregulates *HDAC3* expression, inhibits histone acetylation levels, and thereby reduces transcription of pro-inflammatory factors (IL6 and TNF-α) [274]. H_2_S inhibits *HDAC6* expression, suppresses endothelial dysfunction, and prevents the development of hypertension [275,276]. Sirtuin-1 (SIRT1) is a histone deacetylase. Endogenous H_2_S directly sulfates SIRT1, enhancing the binding of SIRT1 to zinc ions, which then promotes its deacetylation activity and enhances the stability of SIRT1 [277]. Acetyl coenzyme a is important in the performance of the physiological role of acetylases [278,279]. Exogenous H_2_S inhibits the accumulation of acetyl coenzyme a, implying that H_2_S may also be involved in protein acetylation via transcription [280].

### 6.5. H_2_S and Methylation

Methylation is an important form of chromatin remodeling, including DNA and histone methylation. DNA methylation is one of the most important pathways of epigenetic modification. The DNA methylation transferases (DNMTs)-dependent methyl group transfer is the main way in which methylation occurs. Based on the characteristics of their catalytic structural domains, DNMTs in plants are divided into three families, including methyltransferase (MET), chromomethylase (CMT), and domain-rearranged methyltransferases (DRM) [13]. Methylation generally occurs when some cytosine bases are methylated at the 5′ position to become 5-methyl-cytosine (5mC) [270]. There are three types of methylation sites: “CG”, “CHG”, and “CHH” C-base (H for A, C, or T), depending on the sequence of the methylation site. CG methylation relies on DNA methyltransferase 1 (MET1); CHG methylation relies on chromomethylase 3 (CMT3) and CMT2; CHH methylation relies on structural domain rearranged methyltransferase 2 (DR methyltransferase 2 (DRM2), CMT2 and CMT3 [270]. There are two regulatory pathways for DNA methylation levels: the family of DNMTs favors methylation levels. DNMT-3a and DNMT-3b are used for ab initio methylation, while DNMT-1 is used to maintain the methylation that is present [281]. Conversely, ten–eleven translocation (Tet) enzymes facilitate the removal of 5-methylcytosine [281]. Similarly, demethylases play similar roles, including DEMETER (DME) and the Repressor of Silencing 1 (ROS1) [13]. DNA, which is methylation-dependent on the activity of DNMTs, was based on *S*-adenosyl-l-methionine (SAM) as the methyl donor. Similarly, histone methylation is also dependent on the activities of the histone methyltransferase (HMT) family, mainly including histone lysine methyltransferases (HKMTs) and protein arginine methyltransferases (PRMTs), with SAM as the methyl donor [270].

H_2_S is closely associated with methylation phenomena [13,282,283,284] (Figure 6). Osmotic stress induces the production of endogenous H_2_S in *Setaria italica* L., accompanied by the upregulation of DNMTs activity, causing the expression of drought-resistant TFs (*AREB1*, *DREB2A*, *ZIP44*, *NAC5* expression) and ultimately increasing the resistance to stress [13]. Pretreatment with 2 μmol L^−1^ NaHS significantly inhibited ETH release and pectin synthesis, increased pectin methylation, and reduced the Al accumulation in the cell wall in rice [282]. Methylation at protein arginine methyltransferase 5 (PRMT5) in *Arabidopsis* increased the enzymatic activity of AtLCD, thereby enhancing the endogenous H_2_S signaling and improving Cd^2+^ tolerance [283]. These findings suggest a potential link between H_2_S and DNA methylation.

## 7. Conclusions and Prospects

The pathways for the synthesis and metabolism of biologically endogenous H_2_S are summarized in Section 2. At present, H_2_S generation by the DAO/3-MST pathway has only been found in animals, and it is worthy of further study to establish whether a similar pathway exists in plants.

With the function of H_2_S in plants gradually being revealed, its donor has been used extensively to reveal a variety of roles in the plant. H_2_S studies are all based on pharmacological experiments that are carried out through the use of combinations of H_2_S donors, inhibitors, and scavengers. It is important to select the right H_2_S donor, depending on the purpose of the study. In this paper, the selection of H_2_S donors and inhibitors (scavengers) was summarized. The classical donors of Na_2_S, NaHS, GYY4137, thioglycine, and TV are more widely used in plant studies. Research at the subcellular level is inevitable in the future, but it is clear that the substances mentioned above are not appropriate. AP39, AP123, and RT01 represent H_2_S donors that specifically target mitochondria and are more suited to the relevant cellular-level studies. If H_2_S is to be used in practical production, the donor needs to be water-soluble, stable in action, highly controllable, and non-toxic. Conventional H_2_S donors are equally unsuitable. The new environmentally friendly hydrogen sulfide-releasing agents, represented by dialkyldithiophosphates, can be a good solution. However, these substances are not very suitable for facility cultivation (hydroponics). It is evident that the research and development of new H_2_S donors still need to be enhanced. The concentration of endogenous H_2_S is very important for the physiological regulation of plants; finding the means to accurately determine the amount of H_2_S in plants is a problem that needs to be overcome in the relevant studies. In terms of the functional impact of H_2_S on proteins, the basic process generally follows the biotin conversion method. The N-ethyl maleimide method is the most commercially successful method. Unlike proteomic studies, organelle studies focus more on in vivo-specific localization. The study of fluorescent probes for H_2_S provides technical support for research in this direction. The optimization of measurement methods and techniques has a positive effect on the research.

H_2_S plays an important role as a gaseous transmitter throughout the life cycle of plants. The persistence of plants (sessile organisms) in situ leads to stress tolerance becoming necessary for plants to face adversity. Numerous reports have shown that H_2_S has a positive impact on improving plant stress tolerance. However, the topic deserves further study. For example, H_2_S crosstalk exists with other signaling molecules (e.g., GTs), plant hormones, and PGRs. Exactly how they regulate one another and also their functions remain to be determined. In particular, the precise determination of the location of H_2_S in the signaling pathway is a crying need, and the direction needs to be verified in a large number of future experiments. Numerous experiments have now confirmed that H_2_S has a positive effect on the improvement of plant stress resistance, and the main mode of its regulatory action is PTMs, but most of the studies on this modifying mode in plants are related to phenotypic traits that are visible to the naked eye (e.g., wilting or senescence). The mechanism of the cascade crosstalk relationship between this modification mode and other signaling molecules (e.g., NO and CO) at the cellular and subcellular levels needs to be further clarified to provide a reference for elucidating the mechanism of action studies in the future.

## Figures and Tables

**Figure 1 ijms-23-15107-f001:**
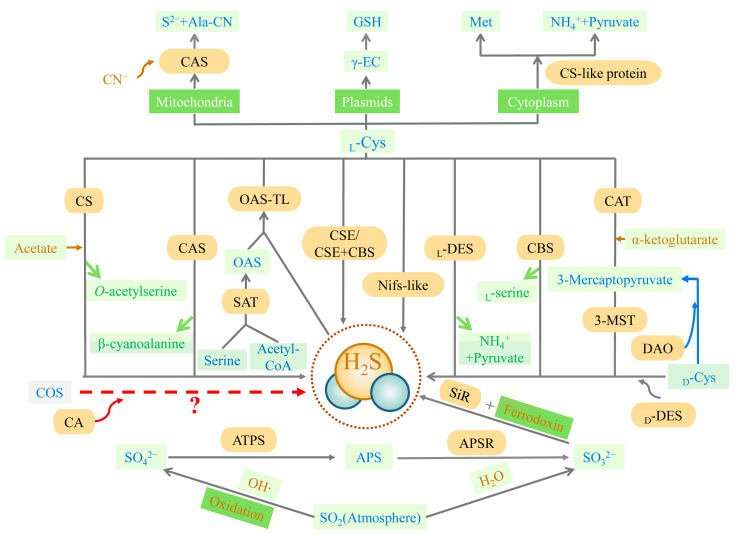
Synthesis of H_2_S. APS (adenosine 5′-phosphosulfate), APSR (APS reductase, EC 1.8.99.2), ATPS (ATP sulfurylase, EC 2.7.7.4), CA (carbonic anhydrase, EC 4.2.1.1), CAS (l-3-cyanoalanine synthase, EC 4.4.1.9), CAT (cysteine aminotransferase, EC 2.6.1.3), CBS (cystathionine-β-synthase, EC 4.2.1.22), COS (carbonyl sulfide), CN^−^ (cyanide), CS (*O*-acetyl-l-serine via cysteine synthase, EC 4.2.99.8), CSC (hetero-oligomeric cysteine synthase complex), CSE (cystathionine-γ-lyase, EC 4.4.1.1), DAO (d-amino acid oxidase, EC 1.4.3.3), d-Cys (d-cysteine), d-DES (d-cysteine desulfhydrase, EC 4.4.1.15), γ-EC (γ-glutamylcysteine), GSH (glutathione), HO· (hydroxyl radicals), H_2_O (water), H_2_S (hydrogen sulfide), l-Cys (l-cysteine), l-DES (l-cysteine desulfhydrase, EC 4.4.1.1), Met (methionine), 3-MST (3-mercaptopyruvate sulfurtransferase, EC 2.8.1.2), NH_4_^+^ (ammonium), Nifs-like (nitrogenase Fe-S cluster-like), OAS (*O*-acetyl-l-serine), OSATL (*O*-acetylserine(thiol)lyase, EC 2.5.1.47), S^2−^ (Sulfide), SAT (serine acetyltransferase, EC 2.2.1.30), SiR (sulfite reductase, EC 1.8.7.1), SO_2_ (sulfur dioxide), SO_3_^2−^ (sulfite), SO_4_^2−^ (sulfate). The closed lines filled with yellow represent enzymes. Rectangles represent substance/organelle reaction conditions. Rectangles filled with light green represent substances (wherein the blue-lettered cells are pivotal substances, yellow-lettered cells are involved in the synthesis of substrates, and green-lettered cells are reaction synthesis byproducts). Rectangles filled with dark green represent substance organelles/reaction conditions (wherein the white-lettered cells are organelles and the yellow-lettered cells are reaction conditions). Yellow arrows represent the substrates involved in the reaction, green arrows represent byproducts of the reaction, and red arrows represent the need for verification by further research. The reactions involved in the blue arrows are not currently found in plants.

**Figure 2 ijms-23-15107-f002:**
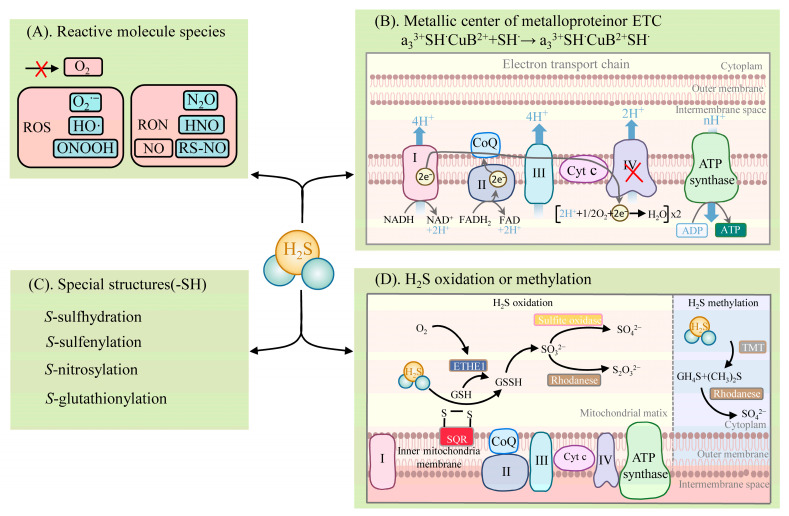
Metabolism of H_2_S: (**A**) reaction of H_2_S with reactive molecule species; (**B**) binding or electron transfer of H_2_S to the metal center of a metalloprotein; (**C**) reaction of H_2_S with proteins with specific structures (involved in post-translational modifications); (**D**) metabolic pathways of H_2_S oxidation and methylation. Note: I (mitochondrial complex I), II (mitochondrial complex II), III (mitochondrial complex III), IV (mitochondrial complex IV), Cyt c (Cytochrome c), CoQ (coenzyme Q/ubiquinone), CH_4_S (methanethiol), (CH_3_)_2_S (dimethyl sulfide), ETHE1 (ethylmalonic encephalopathy 1 protein), GSH (glutathione), GSSG (glutathiol), HNO (nitroxyl), H_2_S (hydrogen sulfide), HO· (hydroxyl radicals), N_2_O (nitrous oxide), NO (nitric oxide), O_2_ (oxygen), O_2_^•−^ (superoxide radical), ONOOH (peroxynitrite), RON (reactive nitrogen species), ROS (reactive oxygen species), RS-NO (*S*-nitrosothiols), SO_4_^2−^ (sulfate), S_2_O_3_^2−^ (thiosulfuric acid), SO_3_^2−^ (sulfite), SQR (sulfide quinone reductase), TMT (thiol *S*-methyl-transferase), TST (thiosulfate sulfurtransferase). A red cross indicates an inhibitory effect on enzyme activity or the production of a substance.

**Figure 3 ijms-23-15107-f003:**
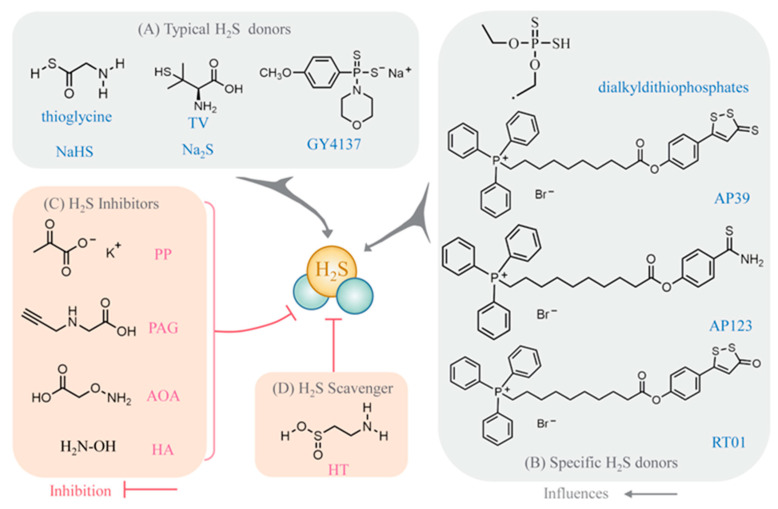
The donors of H_2_S (inhibitor (scavenger)): (**A**) Traditional H_2_S donor; (**B**) special H_2_S donor; (**C**) inhibitor of H_2_S; (**D**) scavenging of H_2_S. Note: AP39 (anethole dithiolethione/(10-oxo-10-(4-(3-thioxo-3H-1,2-dithiol-5yl)phenoxy)decyl) triphenylphosphonium bromide), AP123 (hydroxythiobenzamide), AOA (aminooxyacetic acid), GY4137 (morpholin-4-ium 4-methoxyphenyl (morpholino) phosphinodithioate), HA (hydroxylamine), HT (hypotaurine), NaHS (sodium hydrosulfide), Na_2_S (sodium sulfide), PAG (dl-propargylglycine), PP (potassium pyruvate), RT01 (dialkyldithiophosphates), TV (l-thiovaline).

**Figure 4 ijms-23-15107-f004:**
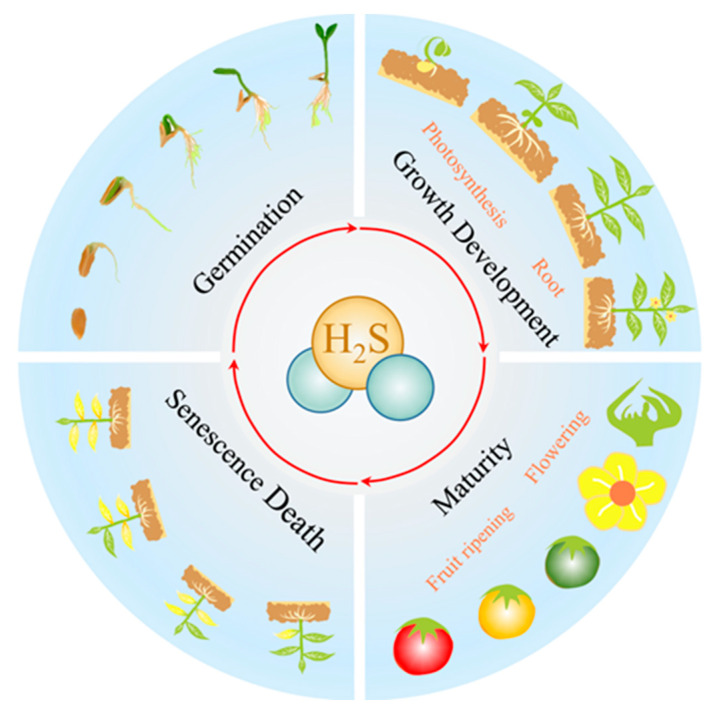
H_2_S is involved in the growth and development of plants.

**Figure 5 ijms-23-15107-f005:**
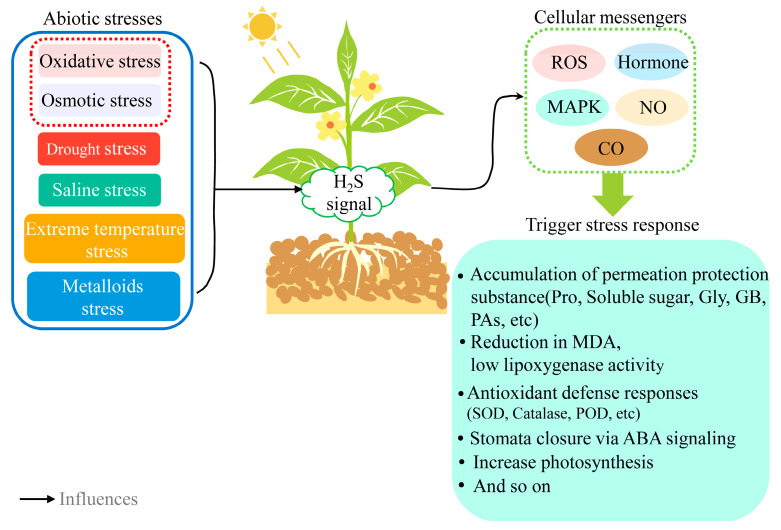
The effect of H_2_S on plant stress tolerance. ABA (abscisic acid), CO (carbon monoxide), GB (glycine betaine), Gly (glycine), H_2_S (hydrogen sulfide), MAPK (mitogen-activated protein kinase), MDA (malondialdehyde), PAs (polyamines), POD (peroxidase), Pro (proline), ROS (reactive oxygen species), SOD (superoxide dismutase).

**Figure 6 ijms-23-15107-f006:**
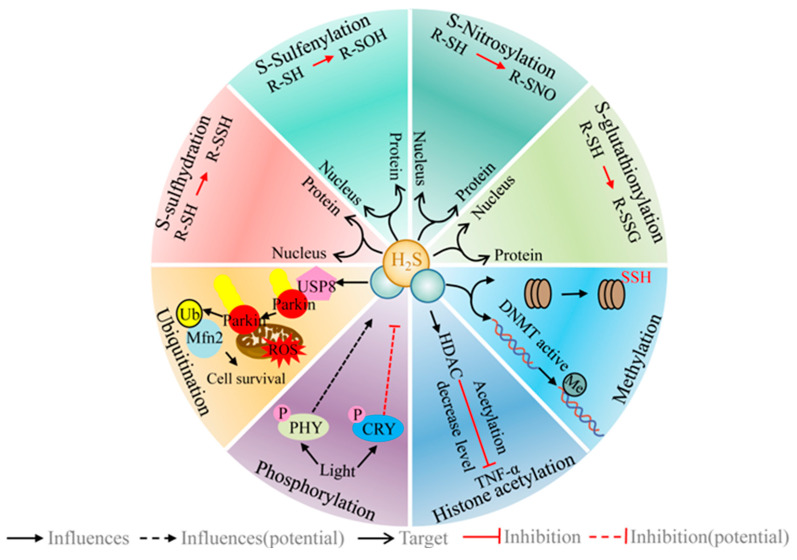
H_2_S regulates a range of different PTMs/signaling systems. CRY (cryptochromes), DNMTs (DNA methylation transferases), HDACs (histone deacetylases), Mfn2 (mitofusin-2), PHY (phytochromes), TNF-α (tumor necrosis factor-α), Ub (ubiquitin), USP8 (ubiquitin-specific peptidase 8).

## Data Availability

Not applicable.

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
