# Peer review of "Biological Functions of Hydrogen Sulfide in Plants"

_ijms, 2022, doi:10.3390/ijms232315107_

Round 1

Reviewer 1 Report

This review present some valuable information regarding the role of hydrogen Sulfide in Plant. Although there are plenty of publication regarding this aspects this review focuses biological role and some physiological aspects of H2S actions.

There are several figures which makes this review easily understandable.

However, it needs some improvement.

Oxidative is vital in regard to H2S effects. So, it should be elaborated. Also, its interaction with NO and other signaling molecules in relation to ROS metabolism must be discussed. For example: https://link.springer.com/article/10.1007/s10725-020-00594-4.

The term 'heavy metal' is rarely used nowadays. Please use toxic metals/metalloids.

I could not see any tables in this paper. Some of the experimental results could be summarized in Tables.

Please cite more recent references.

Author Response

Response letter

Thank you for evaluating our manuscript entitled “Chemical Biology and Biological Function of Hydrogen Sulfide in Plant” (ID: ijms-1995081). These comments provided by you and the reviewers are all valuable and very helpful for revising and improving our paper. We have studied those comments carefully and have made specific corrections, to meet the requirements. The revised portion is marked in blue in the submitted manuscript file with the change-mark. Moreover, we have also adopted some corrections for potential improvement of the manuscript beyond reviewers’ comments. Please review in review mode simple flag state. The main corrections in the paper and the response to the reviewer’s comments are as follows:

Responses to the comments of Reviewer #1

Comment (1): Oxidative is vital in regard to H2S effects. So, it should be elaborated. Also, its interaction with NO and other signaling molecules in relation to ROS metabolism must be discussed. For example: https://link.springer.com/article/10.1007/s10725-020-00594-4.

Response 1: Thank you for the suggestion. We have added the discussion in the “5.1 H2S and oxidative stress” section (page 15, lines 565-602).

Changes in ROS levels in cells can alter the structure and function of a variety of proteins, thereby affecting many different signal transduction pathways. Under stress, interactions between ROS production and scavenging in different compartments of plants can generate specificity [151]. Among them, Peroxisomes are subcellular compartments important for ROS metabolism in plants. In plant peroxisomes, xanthine oxidoreductase (XOR) activity produces uric acid, which is accompanied by the production of O2•-. NOS-like generates NO by using L-arginine and Ca2+ as substrate activity. NO reacts with O2•- thus converting to the less toxic ONOO- [152]. H2O2-producing enzymes (Catalase, GOX) are regulated by NO-mediated S-nitrosation and tyrosine nitration [153]. The O2•- can also be converted to H2O2 by CuZnSOD. In addition, glycolate oxidase (GOX), acyl-CoA oxidase (AOX), urate oxidase (UO), polyamine oxidase, copperamine oxidase (CuAO), sulfite oxidase (SO), and sarcosine oxidase (SOX) are important sources of the peroxisomal H2O2 pool. In the peroxisome, Catalase located in the substrate works with APX located in the membrane to break down H2O2 [152]. In plants, both Catalase [154] and GOX [155] activities can be inhibited by H2S. Studies have found that H2S is present in plant peroxisomes [154]. Evidently, H2S and NO play an important role in the metabolism of ROS in the peroxisome. Antioxidant enzymes play an important role in ROS scavenging. Proteomics studies have shown that anti-oxidant enzymes are important targets of H2S-mediated S-sulfhydration, including matrix sAPX and cystoid thylakoid tAPX, Catalase 1/2/3, DHAR1/2/3, GR1/2, etc [156]. Chen et al. speculated that H2S may interact with ROS homeostasis by acting against oxidases and/or other targets in the signaling pathway [157]. From the above, it is clear that H2S is involved in all ROS metabolism.

The interaction between H2S and other signaling molecules jointly regulates ROS. Both NO and H2S enhance plant antioxidant capacity by reducing excess production of ROS, decreasing lipid peroxidation and increasing the activity of major antioxidant enzymes [158]. Pharmacological experiments showed that H2S donors increased NO content and vice versa, accompanied by upregulation of several antioxidant enzyme activities (e.g. SOD, Catalase and APX) [159]. In maize, 0.2 mmol L-1 SNP (NO donor) promotes the activity of endogenous H2S synthases (L-DES, CAS and CS) and H2S ac-cumulation [160]. Similarly, NO was observed in tomato to stimulate H2S accumulation by upregulating the transcript levels of H2S biosynthetic enzymes and to act down-stream of NO to reduce oxidative stress [161]. Exogenous H2S reduced NO levels and controlled salinity-induced oxidative and nitrosative cell damage [162]. Carbon monoxide (CO) has been shown to protect plants from the harmful effects of excess ROS [163]. Heme oxygenase-1 (HO-1) is an important source of endogenous CO in plants [164]. In pepper, NaHS induces CsHO-1 gene and CsHO-1 protein expression in a time-dependent manner [165]. Hemoglobin (CO scavenger) treatment enhanced H2S biosynthesis (L-cysteine desulfhydrase, L-DES), which induced the accumulation of H2S in tobacco cells [166].

Comment (2): The term 'heavy metal' is rarely used nowadays. Please use toxic metals/metalloids.

Response 2: We are so sorry for the mistake and very grateful for your advice, and we have changed “heavy metal” to “metalloids” in the revised manuscript (page 1-19, lines 40, 544, 794, 795, 798, 801, 803 and 809).

Figure 5. The effect of H2S on plant stress tolerance. ABA (abscisic acid), CO (carbon monoxide), GB (glycine betaine), Gly (glycine), H2S (Hydrogen sulfide), MAPK (Mitogen-activated protein kinase), MDA (malondialdehyde), PAs (polyamines), POD (peroxidase), Pro (Proline), ROS (reactive oxygen species), SOD (superoxide dismutase).

Comment (3): I could not see any tables in this paper. Some of the experimental results could be summarized in Tables.

Response 3: Thank you for the suggestion. We have added the tables in the “4. H2S is involved in plant growth and development” (page 13) and “5. H2S improves plant stress tolerance” sections (pages 20).

Table 1. H2S is involved in plant growth and development.

Development

Plant species

H2S doses

Effects

References

Germination

Arabidopsis thaliana

12 μmol L-1 NaHS

Activated AOX

Mediated cyanide-resistant respiration pathway

[85]

Arabidopsis thaliana

0.1 mmol L-1 NaHS

Maintained protein stability of ABSCISIC ACID-INSENSITIVE 4

[84]

Root development

Peach (Prunus persica (L.) Batsch)

0.2 mmol L-1 NaHS

Increased the concentration of endogenous auxin

Up-regulated the expression of LATERAL ORGAN BOUNDARIES DOMAIN 16

[148]

Cucumber (Cucumis sativus L.)

100 μmol L-1 NaHS

Promoted the occurrence of adventitious roots

Regulated osmotic substance content (proline) and enhanced antioxidant ability

[108]

Photosynthesis

Wheat (Triticum aestivum L.)

200 µmol L-1 NaHS

Increased the photosynthesis and carbohydrate metabolism

[111]

Garlic (Allium sativum)

200 µmol L-1 NaHS

Increased the values of net photosynthetic rate, transpiration rate and stomatal conductance

[110]

Flowering

Heading Chinese cabbage (B. rapa L. syn. B. campestris L. ssp. chinensis Makino var. pekinensis (Rupr.)

J. Cao et Sh. Cao)

100 µmol L-1 NaHS

Promoted plant flowering by weakening or eliminating the binding abilities of BraFLCs to downstream promoters through S-sulfhydration

[122]

Maturity and senescence

Strawberry (Fragaria chiloensis (L.) Mill.)

0.2 mmol L-1 NaHS

Maintained fruit firmness

Delayed pectin degradation

Downregulated the expression of polygalacturonase, pectate lyase, and expansin

[132]

Goji berry (Lycium barbarum L.)

1.4 mmol L-1 NaHS

Improved postharvest quality, Increased the bioactive compounds accumulation

Boosted the antioxidant capacity

[130]

NaHS: sodium hydrosulfide; AOX: alternative oxidase.

Table 2. H2S improves plant stress tolerance.

Plant species

Stressors

H2S doses

Protective effects

References

Arabidopsis thaliana

Oxidative

0.5 mmol L-1 NaHS

Repressed glycolate oxidase

activities

[155]

Arabidopsis thaliana

Osmotic

150 mmol L-1 NaHS

Involved in osmotic

stress-triggered stomatal closure

[171]

Safflower

Drought: 70 and 50% field capacity

0.5 and 1.0 mmol L-1 NaHS

Increased the accumulation of secondary metabolites

Strengthened the antioxidant capacity

Regulated elemental uptake

[174]

Wheat (Triticum aestivum L.)

Drought: 30% field capacity

10 mg m-3 SO2

Triggered proline accumulation

Activated antioxidant enzymes

Changed expression level of transcription factors

Increased H2S content

[175]

Cyclocarya paliurus

Salinity: 100 mmol L-1

NaCl

0.5 mmol L-1 NaHS

Maintained chlorophyll fluorescence

Regulating nitric oxide level

Improved antioxidant capacity

[196]

Wheat (Triticum aestivum L.)

Heat: 40 ℃

200 µmol L-1 NaHS

Reduced glucose sensitivity

Increased the activities of SOD, CAT and AsA-GSH cycle

[213]

Pepper (Capsicum annuum L.)

Chilling: 10 ℃/5 ℃ day/night

1 mmol L-1 NaHS

Enhanced the antioxidant capacity

Increased the enzyme transcription levels

Reduced the contents of O2•−, H2O2, and MDA

[212]

Wheat (Triticum aestivum L.)

Rice

(Oryza sativa L. var.)

Metalloids: 20 μmol L-1 Cr(VI)

15 μmol L-1 NaHS

Maintained fruit firmness

Delayed pectin degradation

Downregulated the expression of polygalacturonase, pectate lyase, and expansin

[224]

SOD: Superoxide dismutase; CAT: Catalase; AsA-GSH: ascorbate-glutathione cycle; MDA: malondialdehyde; O2•−: superoxide radical; H2O2: hydrogen peroxide.

Comment (4): Please cite more recent references.

Response 4: References were replaced with more recent studies (pages 27-39, lines 1140-1820).

Thank you very much.

Sincerely,

Shuhua Zhu, PhD

Agricultural College, Shandong Agricultural University,

Taian, Shandong, 271018, China

Email:shuhua@sdau.edu.cn

Reviewer 2 Report

The manuscript entitled " Chemical Biology and Biological Function of Hydrogen Sulfide in Plant" by Zhifeng Yang et al., try to comprehensively review the physiological function and biochemical characteristics of H2S in plants. This is a very systematic and valuable work. However, the following questions make me doubt the credibility and scientific rigorous of the manuscript.

 Many concluding sentences lack references. Review articles should respect the original experimental results and not make excessive inferences that are unexpected by the experimental results. There is a lot of redundancies and errors in this paper.

 Authors need to carefully proofread all genes, proteins and words, including italics, abbreviations, capitalization and other issues.

 The author kind introduced the discovery history of H2S in 1st paragraph, which is interesting, but superfluous for the theme of the article.

 Refer to line32-24, the author has too many conclusive statements that lack reference, which is lack of professionalism. The manuscript is riddled with such error.

 Line 39, “Mitochondria are the main sites of endogenous H2S production in plants”. As I know, there is no such conclusion in the original text. Mitochondria are one of the compartments where H2S is produced, but not the “main” one.

 Line 124 “(Li, 2015)”, lost reference.

 Line 119 D-DES, 1st D should be subscript.

 Figure 1 looks complicated because there are too many closed lines. Different colors and shapes are used, but they are not described in the caption.

 Figure2, part A and B both have the problem of mixed use of uppercase and lowercase.

 Line 273 Lack of conjunction “and” between parallel words.

 Line 329-346, there is no clear experiment indicating that H2S retards seed germination in the references. References 87 and 88 seem to illustrate the opposite relationship between H2S and H2O2 in regulating Jatropha Curcas and Vigna radiata germination. How do you explain this contradiction?

In addition, it is not experimental and logical to use the role of H2S and GA in PCD pathway to explain and infer their relationship in seed germination.

 Line 354 “H2S is both positive and inhibitory for seed germination” This sentence is puzzling.

 Line 412 The conclusion described is ambiguous with the conclusion of the article [115]. The original text is as follows: “We found that exogenous H2S-activated ROS production was required for NO generation and that MPK6 mediated H2S-induced NO production. MPK6 was shown to function downstream of ROS and upstream of NO. Finally, we demonstrated that exogenous H2S repressed the distribution of auxin and reduced the meristematic cell division potential in root tips, and NO was involved in this process.”

 Line426 “Broccoli” is wrong. If you want to use latin names, they should be italicized, or common names should be lowercase.

 Line459 “ROS change is the earliest marker event for stomatal closure response.” The "earliest" is lack of evidence. Even the description of H2S and NADPH indicates that H2S regulates stomatal movement earlier than ROS.

 “Suppressor of Overexpression of Constant 1 (SOC I)” SOC I should change to SOC1.

 Line526-528 Co-treatment of H2S and NO”,why don’t you descript the influence of H2S aloneThis deviates from the theme of this paragraph.

 4.4 H2S and senescence

The author had better put the previous description of PCD in this paragraph.

Line 546-550. Based on the background of this paper, H2S in plants, the contents involving other species/human need to be described clearly, especially the full name of genes/proteins mentioned for the first time

 5.1. H2S and oxidative stress

It is recommended to refer to the articles of Corpas, F. J, (10.1111/jipb.12779; 10.3389/fpls.2020.00853; 10.1080/23818107.2019.1619196) and review of Chen et al., (10.1093/jxb/eraa093)

 5.2. H2S and osmotic stress

It is recommended to refer to the articles of 10.3389/fpls.2018.01517

 5.3. H2S and drought stress

SnRK2, RBOH, the name of the genes is incomplete.

The full name is required only when the gene name appears for the first time SnRK2.6 have be mentioned before (Line 465).

The ABA stomatal movement involving H2S is more related to drought stress Previously, the description of stomatal movement involved in the photosynthetic chapter was redundant.

SnRK2.6 is OST1, the name of the same gene needs to be unified.

Ion channels are lack of directional description, for example: K+ channel, K+ in or K+ out?

 Line 671-672 expression of genes, name should be italic

 Line701 L -DES and DCD. The writing method of chiral enzyme needs to be unified. You can refer yourself formulation in line104.

 5.4. H2S and saline stress (SS) You haven't even used the abbreviation of "SS". Why should you mark it.

 Line 736 L -cysteine dehydrogenase should use abbreviation (L-DES)

 Line 814 Persulfation was wrong, you may mean persulfidation.

 6.1.1. H2S and S-sulfhydration

2nd paragraph. Some examples are given, but the description is too complicated. Many conclusions have been mentioned many times before, so it is unnecessary to describe them in detail every time.

 6.3-6.5 If these three chapters lack the research basis in plants, it is suggested to simplify or delete them directly.

 Many of the above problems are not isolated cases, many mistakes are repeated, forgive me for not listing them one by one. 

Author Response

Response letter

Thank you for evaluating our manuscript entitled “Chemical Biology and Biological Function of Hydrogen Sulfide in Plant” (ID: ijms-1995081). These comments provided by you and the reviewers are all valuable and very helpful for revising and improving our paper. We have studied those comments carefully and have made specific corrections, to meet the requirements. The revised portion is marked in blue in the submitted manuscript file with the change-mark. Moreover, we have also adopted some corrections for potential improvement of the manuscript beyond reviewers’ comments. Please review in review mode simple flag state. The main corrections in the paper and the response to the reviewer’s comments are as follows:

Responses to the comments of Reviewer #2

Comment (1): Many concluding sentences lack references. Review articles should respect the original experimental results and not make excessive inferences that are unexpected by the experimental results. There is a lot of redundancies and errors in this paper.

Response 1: Thank you for the suggestion. We have added the relevant references in the revised manuscript.

Comment (2): Authors need to carefully proofread all genes, proteins and words, including italics, abbreviations, capitalization and other issues.

Response 2: We are so sorry for the mistake and very grateful for your advice. We have checked and revised all genes, proteins and words of the whole manuscript.

Comment (3): The author kind introduced the discovery history of H2S in 1st paragraph, which is interesting, but superfluous for the theme of the article.

Response 3: We are very grateful for the advice. We have deleted the first paragraph.

Comment (4): Refer to line32-24, the author has too many conclusive statements that lack reference, which is lack of professionalism. The manuscript is riddled with such error.

Response 4: Thank you for the suggestion. The first paragraph has been deleted in line with the suggestion in comment (3). We have added the relevant references in the revised manuscript.

Comment (5): Line 39, “Mitochondria are the main sites of endogenous H2S production in plants”. As I know, there is no such conclusion in the original text. Mitochondria are one of the compartments where H2S is produced, but not the “main” one.

Response 5: As suggested, we have revised the sentence “Mitochondria are the main sites of endogenous H2S production in plants” to “Mitochondria are one of the compartments where H2S is produced” (page 1, lines 31).

Comment (6): Line 124 “(Li, 2015)”, lost reference.

Response 6: We are very grateful for the advice. we have revised the sentence “Atmospheric SO2 can also produce SO32- spontaneously by non-enzymatic interaction with H2O (Li, 2015).” to “Atmospheric SO2 can also produce SO32- spontaneously by non-enzymatic interaction with water”. We have deleted this reference (page 3-4, lines 122-123).

Comment (7): Line 119 D-DES, 1st D should be subscript.

Response 7: We are so sorry for the mistake and very grateful for your advice. We have checked and revised the abbreviations of the whole manuscript. We’ve changed “D-DES” to “D-DES” throughout the manuscript.

Comment (8): Figure 1 looks complicated because there are too many closed lines. Different colors and shapes are used, but they are not described in the caption.

Response 8: We are very grateful for the advice. We have supplemented the graphical interpretation of Figure 1 (pages 3, lines 101-109).

Figure 1. Synthesis of H2S. APS (adenosine 5’-phosphosulfate), APSR (APS reductase, EC 1.8.99.2), ATPS (ATP sulfurylase, EC 2.7.7.4), CA (carbonic anhydrase, EC 4.2.1.1), CAS (L-3-cyanoalanine synthase, EC 4.4.1.9), CAT (cysteine aminotransferase, EC 2.6.1.3), CBS (cystathionine-β-synthase, EC 4.2.1.22), CN- (cyanide), CS (O-acetyl-L-serine via cysteine synthase, EC 4.2.99.8), CSC (hetero-oligomeric cysteine synthase complex), CSE (cystathionine-γ-lyase, EC 4.4.1.1), DAO (D-amino acid oxidase, EC 1.4.3.3), D-Cys (D-cysteine), D-DES (D-cysteine desulfhydrase, EC 4.4.1.15), γ-EC (γ-glutamylcysteine), GSH (glutathione), HO· (hydroxyl radicals), H2O (water), H2S (hydrogen sulfide), L-Cys (L-cysteine), L-DES (L-cysteine desulfhydrase, EC 4.4.1.1), Met (methionine), 3-MST (3-mercaptopyruvate sulfurtransferase, EC 2.8.1.2), NH4+ (ammonium), Nifs-like (nitrogenase Fe-S cluster-like), OAS (O-acetyl-L-serine), OSATL (O-acetylserine(thiol)lyase, EC 2.5.1.47), S2- (Sulfide), SAT (serine acetyltransferase, EC 2.2.1.30), SiR (sulfite reductase, EC 1.8.7.1), SO2 (sulfur dioxide), SO32- (sulfite), SO42- (sulfate). The closed lines filled with yellow represent enzymes. Rectangles represent substances/ organelles/ reaction conditions. Rectangles filled with light green represent substances (where blue lettered cells are pivotal substances, yellow lettered cells are involved in the synthesis of substrates and green lettered are reaction synthesis by-products). Rectangles filled with dark green represent substances organelles/reaction conditions (where the white lettered cells are organelles and the yellow lettered cells are reaction conditions). Yellow arrows represent substrates involved in the reaction, green arrows represent byproducts of the reaction, red arrows represent to be verified by further research, and blue arrows represent not currently found on plants.

Comment (9): Figure2, part A and B both have the problem of mixed use of uppercase and lowercase.

Response 9: We are very grateful for the advice. We have changed the case of the text in Figure 2 (page 5).

Figure 2. Metabolism of H2S. (A) Reaction of H2S with reactive molecule species. (B) Binding or electron transfer of H2S to the metal center of a metalloprotein. (C) Reaction of H2S with proteins with specific structures (involved in post-translational modifications). (D) Metabolic pathways of H2S oxidation and methylation. I (mitochondrial complex I), II (mitochondrial complex II), III (mitochondrial complex III), Ⅳ (mitochondrial complex Ⅳ), Cyt c (Cytochrome c), CoQ (coenzyme Q/ubiquinone), CH4S (methanethiol), (CH3)2S (dimethyl sulfide), ETHE1 (ethylmalonic encephalopathy 1 protein), GSH (glutathione), GSSG (glutathiol), HNO (nitroxyl), H2S (hydrogen sulfide), HO· (hydroxyl radicals), N2O (nitrous Oxide), NO (nitric oxide), O2 (oxygen), O2•- (superoxide radical), ONOOH (peroxynitrite), RON (reactive nitrogen species), ROS (reactive oxygen species), RS-NO (S-nitrosothiols), SO42- (sulfate), S2O32- (thiosulfuric acid), SO32- (sulfite), SQR (sulfide qui-none reductase), TMT (thiol S-methyl-transferase), TST (thiosulfate sulfurtransferase). A red cross indicates an inhibitory effect on enzyme activity or the production of a substance.

Comment (10): Line 273 Lack of conjunction “and” between parallel words.

Response 10: We are so sorry for the mistake and very grateful for your advice. We have revised “This method effectively eliminates the effect of intramolecular disulfides intermolecular disulfides” to “This method effectively eliminates the effect of intramolecular disulfides and intermolecular disulfides” in the revised manuscript (page 8, line 276-277).

Comment (11): Line 329-346, there is no clear experiment indicating that H2S retards seed germination in the references. References 87 and 88 seem to illustrate the opposite relationship between H2S and H2O2 in regulating Jatropha Curcas and Vigna radiata germination. How do you explain this contradiction?

Response 11: We are very grateful for the advice. The sentence is to describe that H2S delays seed germination in a dose-dependent manner (Page 8, Line 319-320).

Comment (12): In addition, it is not experimental and logical to use the role of H2S and GA in PCD pathway to explain and infer their relationship in seed germination.

Response 12: As suggested, we have deleted part of the description, combined with the recommendations in Comment (19), we have moved this part “the role of H2S and GA in the PCD pathway” to section “4.4. H2S and senescence” (page 12-13, line 506-520).

Comment (13): Line 354 “H2S is both positive and inhibitory for seed germination” This sentence is puzzling.

Response 13: We are sorry for the confusion caused by inaccurate language. The point we are trying to make is that most studies have shown that H2S has a positive effect on seed germination, especially under stress conditions (DOI: 10.1080/03650340.2021.1912321; 10.3389/fpls.2018.01288; 10.1016/j.plaphy.2022.09.003). However, some studies have also found that H2S has an inhibitory effect on Arabidopsis seed germination (10.3390/ijms23031389; 10.1515/biolog-2015-0083; 10.1007/s11738-012-1021-z). We have revised the sentence “H2S is both positive and inhibitory for seed germination” to “Low concentrations of H2S promote seed germination while high concentrations of H2S inhibit seed germination” in the revised manuscript (page 9, lines 333-334).

Comment (14): Line 412 The conclusion described is ambiguous with the conclusion of the article [115]. The original text is as follows: “We found that exogenous H2S-activated ROS production was required for NO generation and that MPK6 mediated H2S-induced NO production. MPK6 was shown to function downstream of ROS and upstream of NO. Finally, we demonstrated that exogenous H2S repressed the distribution of auxin and reduced the meristematic cell division potential in root tips, and NO was involved in this process.”

Response 14: We are very grateful for your advice. We have rewritten the conclusion to make the description clearer and more explicit. According to the mechanism diagram in the original text, it is described as follows: High concentrations of H2S activated the ROS-(MITOGEN-ACTIVATED PROTEIN KINASE 6) MPK6-NO signaling pathway and inhibited primary root (PR) growth. During this process, ROS production activated by exogenous H2S is required for NO generation, and MPK6 mediates H2S-induced NO production (page 10, line 394-398).

Comment (15): Line426 “Broccoli” is wrong. If you want to use latin names, they should be italicized, or common names should be lowercase.

Response 15: We are so sorry for the mistake and very grateful for your advice. We have revised the word “broccoli” lowercased (page 11, line 410).

Comment (16): Line459 “ROS change is the earliest marker event for stomatal closure response.” The "earliest" is lack of evidence. Even the description of H2S and NADPH indicates that H2S regulates stomatal movement earlier than ROS.

Response 16: We are so sorry for the mistake and very grateful for your advice, and we have revised “ROS change is the earliest marker event for stomatal closure response” to “ROS changes are early marker events for the stomatal closure response” in the revised manuscript (page 11, lines 427-428).

Comment (17): Suppressor of Overexpression of Constant 1 (SOC I)” SOC I should change to SOC1.

Response 17: As suggested, we have revised “SOC I” to “SOC 1” (page 11, line 449-452).

Comment (18): Line526-528 “Co-treatment of H2S and NO”why don’t you descript the influence of H2S aloneThis deviates from the theme of this paragraph.

Response 18: We are very grateful for your suggestion. We have removed the co-processing. We have added a description of H2S (page 12, lines 495-498). Added as follows: In strawberry, exogenous H2S significantly inhibited softening by decreasing the activities of PMG, PG and EGase [131]. Similarly, in Chilean strawberry, 0.2 mmol L-1 NaHS could reduce the expression of pectin degradation-related genes (FcPG1, FcPL1, FcEXP2, FcXTH1, FcEG1) and alleviate pectin degradation, thereby prolonging fruit shelf life [132].

Comment (19): 4.4 H2S and senescence The author had better put the previous description of PCD in this paragraph.

Response 19: As suggested, we have moved the description of PCD in the “4.4 H2S and senescence” section (page 12-13, line 506-520).

Comment (20): Line 546-550. Based on the background of this paper, H2S in plants, the contents involving other species/human need to be described clearly, especially the full name of genes/proteins mentioned for the first time

Response 20: Thank you for the suggestion. We have added the full name of genes/proteins in the revised manuscript.

Comment (21): 5.1. H2S and oxidative stress. It is recommended to refer to the articles of Corpas, F. J, (10.1111/jipb.12779; 10.3389/fpls.2020.00853; 10.1080/23818107.2019.1619196) and review of Chen et al., (10.1093/jxb/eraa093)

Response 21: We are very grateful for the advice. According to the reference (10.1111/jipb.12779; 10.3389/fpls.2020.00853; 10.1080/23818107.2019.1619196 and 10.1093/jxb/eraa093), we have added the description in the “5.1. H2S and oxidative stress” section (pages 15, line 565-602).

Changes in ROS levels in cells can alter the structure and function of a variety of proteins, thereby affecting many different signal transduction pathways. Under stress, interactions between ROS production and scavenging in different compartments of plants can generate specificity [151]. Among them, Peroxisomes are subcellular compartments important for ROS metabolism in plants. In plant peroxisomes, xanthine oxidoreductase (XOR) activity produces uric acid, which is accompanied by the production of O2•-. NOS-like generates NO by using L-arginine and Ca2+ as substrate activity. NO reacts with O2•- thus converting to the less toxic ONOO- [152]. H2O2-producing enzymes (Catalase, GOX) are regulated by NO-mediated S-nitrosation and tyrosine nitration [153]. The O2•- can also be converted to H2O2 by CuZnSOD. In addition, glycolate oxidase (GOX), acyl-CoA oxidase (AOX), urate oxidase (UO), polyamine oxidase, copperamine oxidase (CuAO), sulfite oxidase (SO), and sarcosine oxidase (SOX) are important sources of the peroxisomal H2O2 pool. In the peroxisome, Catalase located in the substrate works with APX located in the membrane to break down H2O2 [152]. In plants, both Catalase [154] and GOX [155] activities can be inhibited by H2S. Studies have found that H2S is present in plant peroxisomes [154]. Evidently, H2S and NO play an important role in the metabolism of ROS in the peroxisome. Antioxidant enzymes play an important role in ROS scavenging. Proteomics studies have shown that antioxidant enzymes are important targets of H2S-mediated S-sulfhydration, including matrix sAPX and cystoid thylakoid tAPX, Catalase 1/2/3, DHAR1/2/3, GR1/2, etc [156]. Chen et al. speculated that H2S may interact with ROS homeostasis by acting against oxidases and/or other targets in the signaling pathway [157]. From the above, it is clear that H2S is involved in all ROS metabolism.

The interaction between H2S and other signaling molecules jointly regulates ROS. Both NO and H2S enhance plant antioxidant capacity by reducing excess production of ROS, decreasing lipid peroxidation and increasing the activity of major antioxidant enzymes [158]. Pharmacological experiments showed that H2S donors increased NO content and vice versa, accompanied by upregulation of several antioxidant enzyme activities (e.g. SOD, Catalase and APX) [159]. In maize, 0.2 mmol L-1 SNP (NO donor) promotes the activity of endogenous H2S synthases (L-DES, CAS and CS) and H2S accumulation [160]. Similarly, NO was observed in tomato to stimulate H2S accumulation by upregulating the transcript levels of H2S biosynthetic enzymes and to act downstream of NO to reduce oxidative stress [161]. Exogenous H2S reduced NO levels and controlled salinity-induced oxidative and nitrosative cell damage [162]. Carbon monoxide (CO) has been shown to protect plants from the harmful effects of excess ROS [163]. Heme oxygenase-1 (HO-1) is an important source of endogenous CO in plants [164]. In pepper, NaHS induces CsHO-1 gene and CsHO-1 protein expression in a time-dependent manner [165]. Hemoglobin (CO scavenger) treatment enhanced H2S biosynthesis (L-cysteine desulfhydrase, L-DES), which induced the accumulation of H2S in tobacco cells [166].

Comment (22): 5.2. H2S and osmotic stress. It is recommended to refer to the articles of 10.3389/fpls.2018.01517

Response 22: We are very grateful for the advice. According to the reference (10.3389/fpls.2018.01517), we have added the description in the “5.2. H2S and osmotic stress” section (page 16, line 628-636). Add as follows:

ACO is a key enzyme in ethylene biosynthesis in plants. When tomato suffers osmotic stress, ACO is activated and ETH is produced in large quantities. Application of 200 μmol L-1 NaHS and ethylene both induced stomatal closure and water retention. HT supply counteracted the effects of ethylene or osmotic stress on stomatal closure. On the other hand, ethylene-induced H2S negatively regulates ethylene biosynthesis via S-sulfhydration of LeACO1/2. H2S also indirectly inhibited transcription of LeACO1 and LeACO2. This showed that endogenous H2S was downstream of osmotic stress signaling. H2S can be involved in osmotic stress signaling through direct or indirect means [120].

(10.3389/fpls.2018.01517)

Comment (23): 5.3. H2S and drought stress. SnRK2, RBOH, the name of the genes is incomplete.

Response 23: As suggested, we have revised “SnRK2” to “Sucrose nonferme-1(SNF1)-RELATED PROTEIN KINASE2 (SnRK2)” (page 17, line 677).

We have revised “respiratory burst oxidase homolog protein (RBOH)” to “e.g., Arabidopsis thaliana respiratory burst oxidase homolog protein D and F (AtROHD and AtROHF)” (page 17, lines 672).

Comment (24): The full name is required only when the gene name appears for the first time SnRK2.6 have be mentioned before (Line 465).

Response 24: As suggested, we have revised “SNF1-RELATED PROTEIN KINASE2.6 (SnRK2.6)” to “SnRK2.6” (lines 50, 432, 685, 1000).

Comment (25): The ABA stomatal movement involving H2S is more related to drought stress Previously, the description of stomatal movement involved in the photosynthetic chapter was redundant.

Response 25: Thank you for the suggestion. We have moved the section on “ABA stomatal movement involving H2S” to the chapter “5.3. H2S and drought stress” and removed the stomatal movement from the chapter on photosynthesis.

Comment (26): SnRK2.6 is OST1, the name of the same gene needs to be unified.

Response 26: Thank you for the suggestion. We have unified the gene name and revised “OST1” to “Open Stomata 1 (OST1)/Sucrose nonferme-1(SNF1)-RELATED PROTEIN KINASE2.6 (SnRK2.6)” (page 2, line 49-50).

Comment (27): Ion channels are lack of directional description, for example: K+ channel, K+ in or K+ out?

Response 27: Thank you for the suggestion. According to the reference (10.1007/s11104-017-3335-5), we’ve changed “In Arabidopsis thaliana, endogenous H2S induces the opening of the K+ channel, which acts as the main osmoregulatory channel in response to drought stress, causing K+ efflux and Ca2+ and Cl- efflux, leading to stomatal closure” to “In Arabidopsis thaliana, endogenous H2S induces the opening of the K+ channel (the outward K+-channel), which acts as the main osmoregulatory channel in response to drought stress, causing K+ efflux and Ca2+ and Cl- efflux, leading to stomatal closure” (page 17, lines 680-683). In addition, according to the reference (0.1111/j.1742-4658.2011.08370.x), we’ve revised “Activation of S-type anion channels (SLAC1), plays a vital role in stomatal closure” to “Activation of S-type anion channels (SLAC1), anion efflux out, plays a vital role in stomatal closure” (page 17, lines 683-685).

[DOI: 10.1007/s11104-017-3335-5] K+ out

[DOI: 10.1111/j.1742-4658.2011.08370.x] anion efflux out

Comment (28): Line 671-672 expression of genes, name should be italic

Response 28: We are so sorry for the mistake and very grateful for your advice, and we have revised “PM H+-ATPase (HvHA1)” to “PM H+-ATPase (HvHA1)” in the revised manuscript (page 17, lines 718).

Comment (29): Line701 L-DES and DCD. The writing method of chiral enzyme needs to be unified. You can refer yourself formulation in line104.

Response 29: We are so sorry for the mistake and very grateful for your advice, we have unified the writing method of chiral enzyme and revised “DCD” to “D-DES” throughout the revised manuscript.

Comment (30): 5.4. H2S and saline stress (SS) You haven't even used the abbreviation of "SS". Why should you mark it.

Response 30: We are so sorry for the mistake and very grateful for your advice, we have deleted this abbreviation.

Comment (31): Line 736 L-cysteine dehydrogenase should use abbreviation (L-DES)

Response 31: Thank you for the suggestion. We have revised “L-cysteine dehydrogenase” to “L-DES” through the revised manuscript.

Comment (32): Line 814 Persulfation was wrong, you may mean persulfidation.

Response 32: We are so sorry for the mistake and very grateful for your advice, we have revised “persulfation” to “persulfidation” in the revised manuscript (page 21, line 864).

Comment (33): 6.1.1. H2S and S-sulfhydration. 2nd paragraph. Some examples are given, but the description is too complicated. Many conclusions have been mentioned many times before, so it is unnecessary to describe them in detail every time.

Response 33: Thank you for the suggestion. We have simplified this section in the revised manuscript (pages 22, lines 880-904). Revised as follows:

S-sulfhydration is a crucial way in which H2S is involved in life activities [141] (Figure 6). In eukaryotes, cellular autophagy is a highly conserved mechanism of material degradation (the ability to spontaneously eliminate cellular components) [244]. This mechanism is of great significance for differentiation, development, and cell survival. Autophagy-related proteins (ATG) are a group of critical proteins involved in the autophagic process. The H2S target ATG is the most robust evidence for its direct involvement in cellular autophagy. In animals, H2S modifies the Cys150 residue of GAPDH by S-sulfhydration to achieve regulation of cellular autophagy. Similarly, in plants, H2S can modify residues Cys170 of ATG4 and Cys103 of ATG18 via S-sulfhydration to achieve inhibition of cellular autophagy [244]. This mechanism of inhibition of autophagy is independent of redox conditions [245]. The energy sensor Snf1-related protein kinase 1 (SnRK1), the kinase Target of Rapamycin (TOR), ATG1 kinase complex, and the endoplasmic reticulum stress sensor inositol-requiring enzyme-1 (IRE1) are autophagy regulators that have been identified in plants [246]. S-sulfhydration-based modification of these target proteins by H2S is another important way they are involved in cellular autophagy [244]. ABA signaling is the classical mechanism that regulates stomatal movement [247]. Similarly, H2S modifies specific cysteine residues through S-sulfhydration, thereby affecting stomatal closure [26,28]. For example, ABI4 [28] and RBOHD [26]. Notably, the complex signaling network of H2S, ABA, and H2O2 cascade crosstalk is an important way for stomatal closure regulatory mechanisms [27,119,183]. In addition, potential target proteins for H2S occur in redox homeostasis, energy status, and cytokinesis-related enzymes, including APX [240], Catalase [154], RBOHD [26], glyceraldehyde 3-phosphate dehydrogen-ase (GAPDH) [240], NADP-isocitrate dehydrogenase (NADP-ICDH) [248], NADP-malic enzyme (NADP-ME) [249], actin [250], et al. It implies that H2S is involved in the above related metabolic activities through S-sulfhydration-based modifications.

Comment (34): 6.3-6.5 If these three chapters lack the research basis in plants, it is suggested to simplify or delete them directly.

Response 34: Thank you for the suggestion. Considering that it may be helpful for the subsequent relevant research work, these parts are retained. We simplified the sections “6.3. H2S and ubiquitination”, “6.4. H2S and histone acetylation” and “6.5. H2S and methylation”. In addition, we supplemented the latest plant work by re-searching the relevant literature in the “H2S and methylation” section (10.1093/plphys/kiac376; 10.1016/j.plaphy.2022.04.006). Revised as follows:

6.3. H2S and Ubiquitination (Ub)

Ub is one of the mechanisms of PTMs prevalent in living organisms, adds ubiquitin to the substrate protein, and thus labels the target protein. Its primary biochemical function is to provide a marker for the subsequent selective degradation of the protein. The ubiquitin-proteasome system (UPS), of which Ub is the core component, is the main pathway for intracellular protein degradation and consists of Ub, ubiquitin-activating enzyme (E1), ubiquitin-conjugating enzyme (E2), ubiquitin-protein ligase (E3), the proteasome and its substrate (protein) [274].

H2S can be involved in regulating protein degradation in cells by modifying ubiquitinated and deubiquitinated components [40,275] (Figure 6). E1 uses the energy generated by ATP hydrolysis to create a thioester bond between the sulfhydryl group of its own catalytic site, cysteine, and the carboxyl group of ubiquitin. The activated ubiquitin is then transferred to the sulfhydryl group of E2. Finally, the transfer of ubiquitin to a specific substrate is carried out by E3. In the UPS, E3 confers specificity to protein degradation reactions. E3 ubiquitin ligases are divided into homologous to the E6-AP carboxyl terminus (HECT)-type, Really Interesting New Gene (RING)-type, and RING-between-RING (RBR)-type [276]. In humans, exogenous H2S modifies ubiquitin-specific peptidase 8 (USP8) via S-sulfhydration to promote USP8-mediated deubiquitination of parkin (an RBR-type E3 ubiquitin ligase) for effective elimination of dysfunctional mitochondria [275]. Similarly, H2S regulates parkin activity and enhances its neuroprotective activity by incorporating bound sulfane sulfur into cysteine residues [40].

6.4. H2S and histone acetylation

The covalent modification of acetyl groups (acetyl groups) from acetyl CoA to the ε-amino group of the N-terminal lysine residue of the histone molecule is called histone acetylation [277].

Mutual antagonistic activity of histone acetyltransferases (HATs) and histone deacetylases (HDACs) is an important way of regulating histone acetylation in plants. HATs catalyze the acetyl transfer of acetyl coenzyme A on histone lysine residues to achieve acetylation [278]. Plant HATs are classified into four categories, including p300/CREB, TATA-binding protein-associated factors, the MYST family protein (MOZ, Ybf2/Sas3, Sas2, and Tip60), and the general control non-repressible 5 (Gcn5)-related N-acetyltransferases (GNATs) [279]. In contrast, HDACs remove the acetyl group from the histone tails and achieve deacetylation, leading to chromatin condensation and reduced gene expression activity [280]. Plant HDACs are divided into three categories, including reduced potassium dependency 3/histone deacetylase 1 (RDP3/HDA1), silent information regulator 2 (SIR2), and histone deacetylase 2 (HD2) [277].

H2S can regulate cellular function by affecting the level of histone acetylation [281-283] (Figure 6). In animals, H2S can regulate cellular function by affecting the acetylation and deacetylation of histones. For example, H2S up-regulates HDAC3 expression, inhibits histone acetylation levels, and thereby reduces transcription of pro-inflammatory factors (IL6 and TNF-α) [281]. H2S inhibits HDAC6 expression, suppresses endothelial dysfunction, and prevents the development of hypertension [282,283]. Sirtuin-1 (SIRT1) is a histone deacetylase. Endogenous H2S directly sulfates SIRT1, enhances the binding of SIRT1 to zinc ions, which then promotes its deacetylation activity and enhances the stability of SIRT1 [284]. Acetyl coenzyme a is important in the performance of the physiological role of acetylases [285,286]. Exogenous H2S inhibits the accumulation of acetyl coenzyme a, implying that H2S may also be involved in protein acetylation via transcription [287].

6.5. H2S and methylation

Methylation is an important form of chromatin remodeling, including DNA and histone methylation. DNA methylation is one of the important pathways of epigenetic modification. DNA methylation transferases (DNMTs)-dependent methyl group transfer is the main way in which methylation occurs. Based on the characteristics of their catalytic structural domains, DNMTs in plants are divided into three families, including methyltransferase (MET), chromomethylase (CMT), and domains rearranged methyltransferases (DRM) [13]. Methylation generally occurs when some cytosine bases are methylated at the 5' position to become 5-methyl-cytosine (5mC) [277]. There are three types of methylation sites: “CG”, “CHG”, and “CHH” C-base (H for A, C, or T) depending on the sequence of the methylation site: CG methylation relies on DNA methyltransferase 1 (MET1); CHG methylation relies on chromomethylase 3 (CMT3) and CMT2; CHH methylation relies on structural domain rearranged methyltransferase 2 (DR methyltransferase 2 (DRM2), CMT2 and CMT3 [277]. There are two regulatory pathways for DNA methylation levels: the family of DNMTs favors methylation levels. DNMT-3a and DNMT-3b are used for ab initio methylation, and DNMT-1 is used to maintain the methylation present [288]. Conversely, ten-eleven translocation (Tet) enzymes facilitate the removal of 5-methylcytosine [288]. Similarly, demethylases play similar roles, including (DEMETER, DME) and Repressor of Silencing 1 (ROS1) [13]. DNA methylation-dependent on the activity of DNMTs is based on S-adenosyl-L-methionine (SAM) as the methyl donor. Similarly, histone methylation is also dependent on the activities of the histone methyltransferase (HMT) family, mainly including histone lysine methyltransferases (HKMTs) and protein arginine methyltransferases (PRMTs), with SAM as the methyl donor [277].

H2S is closely associated with methylation phenomena [13,290-291] (Figure 6). Osmotic stress induces the production of endogenous H2S in Setaria italica L., accompanied by the up-regulation of DNMTs activity, causing the expression of drought-resistant TFs (AREB1, DREB2A, ZIP44, NAC5 expression) and ultimately increasing the resistance to stress [13]. Pretreatment with 2 μmol L-1 NaHS significantly inhibited ETH release and pectin synthesis, increased pectin methylation, and reduced Al accumulation in the cell wall in rice [289]. Methylation at protein arginine methyltransferase 5 (PRMT5) in Arabidopsis increased the enzymatic activity of AtLCD, thereby enhancing endogenous H2S signaling and improving Cd2+ tolerance [290]. These suggest a potential link between H2S and DNA methylation.

Comment (35): Many of the above problems are not isolated cases, many mistakes are repeated, forgive me for not listing them one by one.

Response 35: We are sorry for our negligence and we really appreciate for you careful review. All of these similar mistakes have been corrected throughout the revised manuscript.

Thank you very much.

Sincerely,

Shuhua Zhu, PhD

Agricultural College, Shandong Agricultural University,

Taian, Shandong, 271018, China

Email:shuhua@sdau.edu.cn

Round 2

Reviewer 1 Report

The current version is properly revised.

Author Response

Dear  Reviewer:

Thank you for evaluating our manuscript entitled “Biological Functions of Hydrogen Sulfide in Plant” (ID: ijms-1995081). These comments you and the reviewers provided are valuable and very helpful for revising and improving our paper. We have studied those comments carefully and have made specific corrections to meet the requirements.

Responses to the comments of Reviewer #1

Comments: The current version is properly revised.

Answer: Thank you very much for the valuable comments and constructive suggestions.

Sincerely,

Shuhua Zhu

Reviewer 2 Report

The present manuscript is improved remarkably. A little bit suggestion was shown as below.

The title is a little tedious. How about of “Biological functions of H2S in plant”?

 In the H2S detection methods section, actually includes some methods of detection of PTM -persulfidation. It is better to introduce separately.  

 Some sentences are difficult to understand. For example, line1284-5, “The pathways for the synthesis and metabolism of biologically endogenous H2S are summarized. Whether a similar situation exists in the plant is worthy of further study.”  Since the H2S synthesis pathways have been introduced in the main text, why say that again.  

 Some important literatures are missing. For example, Liu & xue, Plant communication (2021), https://doi.org/10.1016/j.xplc.2021.100179.

Author Response

Dear Reviewer:

Thank you for evaluating our manuscript entitled “Biological Functions of Hydrogen Sulfide in Plant” (ID: ijms-1995081). These comments provided by you and the reviewers are all valuable and very helpful for revising and improving our paper. We have studied those comments carefully and have made specific corrections, to meet the requirements. The revised portion is marked in blue in the submitted manuscript file with the change-mark. Please review in review mode simple flag state. Moreover, we have also adopted some corrections for potential improvement of the manuscript beyond reviewers’ comments. The main corrections in the paper and the response to the reviewer’s comments are as follows:

Responses to the comments of Reviewer #2

Comment (1): The title is a little tedious. How about of “Biological functions of H2S in plant”?

Response 1: Thank you for the suggestion. We have revised the title in the revised manuscript.

Comment (2): In the H2S detection methods section, actually includes some methods of detection of PTM -persulfidation. It is better to introduce separately.

Response 2: We are very grateful for your advice. We have separated the section of H2S detection methods. The methods of detection of PTM-persulfidation was separated in the section “3.3. Detection of protein S-sulfhydration modifications by H2S” (page 8, lines 275-316).

3.3. Detection of protein S-sulfhydration modifications by H2S

Studies of H2S-mediated S-sulfhydration proteins are currently carried out following the biotin conversion approach as the basic process. The primary process has two steps involving the closure of the thiol-blocking reagent and the persulfide by an electrophile reagent and the release of the persulfide by a reducing agent. Specifically, there are four methods commonly used for S-sulfhydration testing. The first method is the classic electrophile S-methyl methanethiosulfonate (MMTS). The MMTS is first used as a sulfhydryl blocker, followed by the labeling of persulfides with N-[6-(biotinamido) hexyl]-3’-(2’-pyridyldithio) propionamide (biotin-HPDP). This method includes S-nitrosation in addition to S-sulfhydration labeling [75]. The second method is that iodoacetic acid (IAA) accurately detects the presence of sulfhydryl groups in the peroxidized proteins. The target groups (-SH, -SSH) in the protein are alkylated by IAA, thus achieving blockage of the free thiol with protein persulfides. DTT specifically cleaves the -SSH group in the protein alkylated. Iodoacetamide-linked biotin (IAP) is labeled against the cleaved moiety by DTT. This method effectively eliminates the effect of intramolecular disulfides intermolecular disulfides [76]. The third method using N-ethyl maleimide is the most widely and commercially used. N-ethyl maleimide, which is linked to both the target group (-SH, -SSH) in the protein, is used as a sulfhydryl group sealer. DTT specifically cleaves the target moiety (-SSH), which has been released by linking Cy5-conjugated maleimide to the site. The in-gel fluorescence signal is reduced in the samples containing the persulfide [77]. The fourth method is the tag-switch test (tag-switch): a two-step reaction involving sulfhydryl blocking (SH-blocking, BR) and tag-switching reagent substitution. The tag-switching reagent contains a reporting molecule (R) and a nucleophile (Nu). Nucleophiles differ in their reactivity towards persulfide adducts due to their reactive properties. The se-lection of suitable nucleophilic reagents for this property allows for the specific detec-tion of disulfide bonds in persulfides. For example, in the first step, methylsulfonyl benzothiazole (MSBT) is used as a sulfhydryl blocking substance for SH-blocking (BR). Next, labeling is completed by binding to labeled tagged-cyanoacetate derivatives, which effectively label persulfides [78]. Recently, a new method for selective persulfide detection, named the Dimedone-Switch method, has been developed, which has the advantages of being specific, rapid, and stable [79]. The method is based on dimedone probes for chemoselective persulfuration labeling. Since protein persulfides (PSSH) are very reactive, their reactivity is similar to that of cysteine residues. Therefore, it be-comes difficult and important to design tools for selective labeling. The Dimedone-Switch method is roughly divided into two steps. In the first step, 4-chloro-7-nitrobenzofurazan (NBF-Cl) treated material was used to block and detect thiols, amines and sulfenic acids. Then, the -S-S-signal detection is based on the dimedone /dimethyl ketone probe. NBF-Cl not only reacts with PSSH and thiols, but also blocks sulfenic acids. NBF-Cl is the key to the specificity of this method for the detection of persulfides. Conventional methods rely on the closure of sulfhydryl groups and persulfides with electrophilic reagents, followed by the release of the latter with reducing agents. These methods require a lot of attention and are more tedious and complex [80].

Comment (3): Some sentences are difficult to understand. For example, line1284-5, “The pathways for the synthesis and metabolism of biologically endogenous H2S are summarized. Whether a similar situation exists in the plant is worthy of further study.” Since the H2S synthesis pathways have been introduced in the main text, why say that again.

Response 3: We apologize for the unclear description and very grateful for the advice. We have revised the sentences “The pathways for the synthesis and metabolism of biologically endogenous H2S are summarized. Whether a similar situation exists in the plant is worthy of further study.” to “The pathways for the synthesis and metabolism of biologically endogenous H2S are summarized in section 2. At present, H2S generation by the DAO/3-MST pathway has only been found in animals, and it is worthy of further study whether a similar pathway exists in plants.” in the revised manuscript (page 26, lines 1087-1090).

Comment (4): Some important literatures are missing. For example, Liu & xue, Plant communication (2021), https://doi.org/10.1016/j.xplc.2021.100179.

Response 4: Thank you for the suggestion. We have added the important literature 178 in sections “5.3 H2S and drought stress” and “6.1.1 H2S and S-sulfhydration” (page 16, lines 651) (page 22, lines 896).

Liu, H.; Xue, S. Interplay between hydrogen sulfide and other signaling molecules in the regulation of guard cell signaling and abiotic/biotic stress response. Plant Commun. 2021, 2, 100179, doi:10.1016/j.xplc.2021.100179.

Thank you very much.

Sincerely,

Shuhua Zhu
